# ON THE GENERALIZATION CAPACITY OF NEURAL NETWORKS DURING GENERIC MULTIMODAL REASONING

**Takuya Ito, Soham Dan, Mattia Rigotti, James Kozloski, & Murray Campbell**
T.J. Watson Research Center, IBM Research
{takuya.ito,soham.dan}@ibm.com, mrg@zurich.ibm.com
{kozloski,mcam}@us.ibm.com

## ABSTRACT

The advent of the Transformer has led to the development of large language models (LLM), which appear to demonstrate human-like capabilities. To assess the generality of this class of models and a variety of other base neural network architectures to multimodal domains, we evaluated and compared their capacity for multimodal generalization. We introduce a multimodal question-answer benchmark to evaluate three specific types of out-of-distribution (OOD) generalization performance: distractor generalization (generalization in the presence of distractors), systematic compositional generalization (generalization to new task permutations), and productive compositional generalization (generalization to more complex tasks structures). We found that across model architectures (e.g., RNNs, Transformers, Perceivers, etc.), models with multiple attention layers, or models that leveraged cross-attention mechanisms between input domains, fared better. Our positive results demonstrate that for multimodal distractor and systematic generalization, either cross-modal attention or models with deeper attention layers are key architectural features required to integrate multimodal inputs. On the other hand, neither of these architectural features led to productive generalization, suggesting fundamental limitations of existing architectures for specific types of multimodal generalization. These results demonstrate the strengths and limitations of specific architectural components underlying modern neural models for multimodal reasoning. Finally, we provide *Generic COG* (gCOG), a configurable benchmark with several multimodal generalization splits, for future studies to explore.

## 1 INTRODUCTION

Recent LLMs appear to exhibit remarkable cognitive abilities. On the surface, these models perform exceptionally well on cognitive tasks, such as language comprehension, mathematical reasoning, and coding (Bubeck et al., 2023; Webb et al., 2023). However, many of these demonstrations are limited to a single modality (e.g., language). Moreover, the mechanisms driving performance among these models are opaque. Even when both the model and dataset are publicly available, the sheer scale of the model and training data make it difficult to isolate which architectural mechanisms influence generalization behavior, in part due to the difficulty in controlling for confounding factors in large pretraining datasets (Kim et al., 2022). To quantify the relationship between mechanism and generalization on multimodal cognitive tasks, we sought to evaluate a set of base neural network architectures (RNN, GRU, Transformer, Perceiver) on a carefully controlled multimodal task.

Recent studies have suggested that the impressive behaviors exhibited by LLMs are due to superficial data interpolation, rather than "emergent cognition" (Schaeffer et al., 2023; Kim et al., 2022; Wu et al., 2023). One approach to adjudicate between possibilities is to first curate carefully controlled training and test sets generated from compositional task experiments (Keysers et al., 2020; Dziri et al., 2023; Bahdanau et al., 2020; Ontañón et al., 2022; Csordás et al., 2022; Hupkes et al., 2020; Lake & Baroni, 2018; Yang et al., 2018; Kim et al., 2022). By design, compositional tasks enable experimenters to measure OOD generalization – the ability to perform tasks beyond the training distribution – by programmatically composing a test set using novel task components. Indeed, when controlling for confounding factors in a training set, studies have shown that neural models cannot

generalize OOD (Kim et al., 2022; Dziri et al., 2023). However, most prior demonstrations have been limited to evaluating tasks within a single modality (e.g., natural language). Thus, it remains unclear to what extent previous unimodal models generalize to multimodal reasoning tasks.

Given recent studies demonstrating that pretraining can actually degrade downstream systematic generalization (Kim et al., 2022), and that OOD generalization performance can inversely scale with model size (McKenzie et al., 2023), we focused on training models from scratch rather than fine-tuning large pretrained models. This ensured full experimental control over training data and architectural choices. We introduce *Generic COG* (gCOG), a task abstracted from the previous COG task (Yang et al., 2018). Our variant employs generic feature sets that are not tied to any specific modality, and relaxes previous experimental constraints to broaden its capacity to test compositional generalization splits (Fig. 1). This design allowed us to comprehensively evaluate a variety of model architectures on tasks that test for three different forms of OOD generalization: 1) Distractor generalization (generalization in the presence of a different noise distribution), 2) Systematic compositional generalization (generalization to new permutations of task structures, i.e., combinatorial generalization), and 3) Productive compositional generalization (generalization to task structures of greater complexity). We find that models that integrate multimodal inputs with either deeper attention layers or cross-attention mechanisms (such as Perceiver-like models) performed best, and were capable of excellent distractor generalization, reasonably good systematic compositional generalization, yet (as with all models tested) no productive compositional generalization. Our results illustrate the successes and fundamental limitations of modern neural architectures' nascent multimodal reasoning abilities, and we provide a configurable multimodal reasoning benchmark for future studies to build upon.

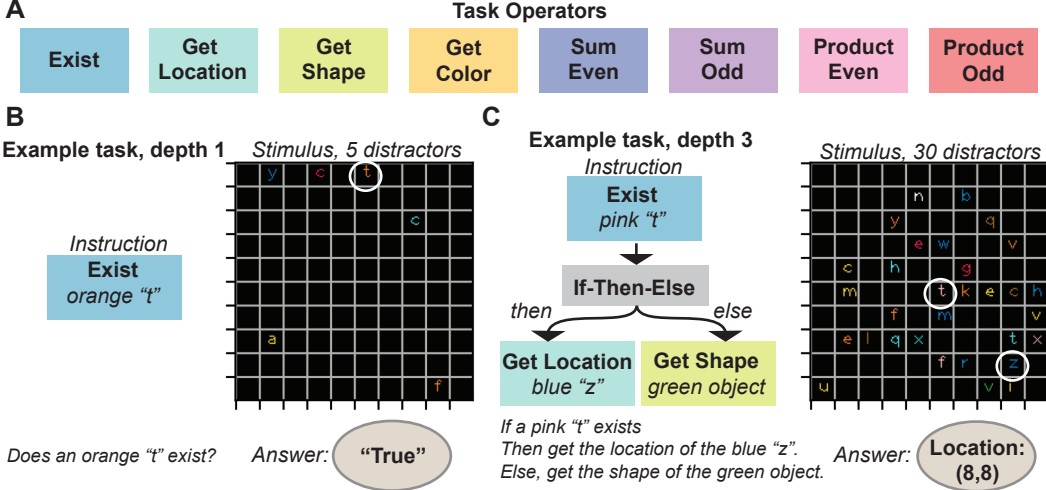

Figure 1: gCOG task. We adapted the previously-developed COG task (Yang et al., 2018). Our modifications to COG included different task operators, the ability to use categorical tokens to allow for generic testing of multimodal reasoning, and the ability to allow for arbitrarily long task instructions to allow for the evaluation of compositional productivity. A) Task operators and objects serve as the core units of the gCOG task. A task operator (e.g., Exist) is paired with a specific feature combination (e.g., orange + "t"). Feature categories correspond to shape (i.e., letters "a" through "z") and color (i.e., 10 discretely coded colors), but can be naturally extended. B) At minimum, a task must comprise of one specific task operator (e.g., Exist) and a feature combination (e.g., orange "t"). An arbitrary number of stimuli (e.g., images) can be constructed on-the-fly to satisfy this task instruction (i.e., produce a TRUE or FALSE response). C) Tasks can be combined with a conditional operator (e.g., an IF-THEN-ELSE conditional) to increase the task complexity. This enables the construction of arbitrarily complex tasks. While the original COG task explored only task trees of depth 3 (i.e., a single conditional), we relaxed this constraint to allow for arbitrarily long task trees. Dataset: https://github.com/IBM/gcog

## 1.1 RELATED WORK

A number of recent studies have evaluated the compositional generalization properties of LLMs, documenting several of their generalization shortcomings (Dziri et al., 2023; Keysers et al., 2020;

Kim et al., 2022; Wu et al., 2023). However, evaluations were limited to massive pretrained language models, making it difficult (if not impossible) to assess the interplay between training data, architecture, and generalization, as noted by Kim et al. (2022). Smaller scale studies demonstrated the limitations of modern neural network architectures (e.g., Transformers, RNNs) on compositional tasks, but have been focused primarily on a single modality (e.g., language) (Csordás et al., 2022; Ontañón et al., 2022; Hupkes et al., 2020; Lake & Baroni, 2018; Andreas, 2019; Bahdanau et al., 2019; Furrer et al., 2021). Other neural models trained on multimodal question-answer datasets like CLEVR (Johnson et al., 2017) also failed to generalize systematically when proper training and test splits were curated (e.g., CLOSURE) (Bahdanau et al., 2020). Related hybrid neuro-symbolic approaches have demonstrated great success at compositional generalization tasks (Klinger et al., 2023; Nye et al., 2020; Bahdanau et al., 2019; Shaw et al., 2021), although these approaches require customized symbolic architectures, possibly rendering them difficult to scale. Extending these studies, our aim is to evaluate the performance of base neural models on generic multimodal reasoning.

Our approach to constructing arbitrarily complex compositions of simple tasks is similar to gSCAN (Ruis et al., 2020). However, it differs in three key ways. First, we focus on question-answer tasks (which require encoder-only architectures), rather than sequence-to-sequence learning tasks (which require decoder architectures, e.g., Qiu et al. (2021)). Sequence decoding tasks introduce the added complexity of requiring autoregressive responses, which are susceptible to fundamental statistical challenges, such as exposure bias (Wang & Sennrich, 2020). Second, gCOG includes a distractor generalization split, in addition to systematic and productive compositional generalization splits. Finally, we methodically characterize different forms of generalization using simpler underlying abstractions (i.e., without the explicit use of image pixels). Indeed, the experimental design is most similar to the original COG (Yang et al., 2018), SQOOP (Bahdanau et al., 2019) and CLOSURE (Bahdanau et al., 2020) in that it is multimodal and can generate synthetic trials on-the-fly. However, those previous tasks did not include productive compositional generalization benchmarks (i.e., evaluation of arbitrarily complex task commands), systematic compositional generalization benchmarks on deeper task trees (e.g., task trees of depth 3), and explicit splits for OOD distractor generalization. In gCOG, we uniquely designed splits that target these three distinct forms of OOD generalization. gCOG therefore provides a scalable design (e.g., more than two modalities can be straightforwardly accommodated) to broadly evaluate multimodal generalization (Fig. 1).

## 1.2 CONTRIBUTIONS

Specific contributions center around the configurability and flexibility of gCOG for three forms of OOD generalization, as well as the comprehensive evaluation of a variety of base neural network architectures on this task. We highlight three principal contributions of this study:

1. A configurable dataset that complements and extends prior tasks (i.e., gSCAN, SQOOP, COG) on multimodal compositional generalization to include productivity splits, systematicity splits on deeper task trees, and OOD distractor generalization splits.

2. A comprehensive evaluation of commonly-used base neural models (RNNs, GRUs, Transformers, Perceivers) on distractor, systematic, and productive generalization splits. We find that for distractor and systematic generalization, including a cross-attention mechanism across input modalities is important. However, all models fail on the productivity split.

3. A comprehensive evaluation of how scaling standard encoder-only Transformer models improves distractor and systematic generalization, but not productive generalization.

Finally, we include analysis of internal model representations in Appendix A.1, revealing the influence of base neural architecture on internal model representations.

## 2 EXPERIMENTAL DESIGN

### 2.1 GCOG FOR MULTIMODAL AND COMPOSITIONAL EVALUATION

gCOG is a configurable question-answer dataset, originally inspired from COG (Yang et al., 2018), that programmatically composes task instructions, and then generates synthetic stimuli to satisfy those instructions on-the-fly (Fig. 1). The primary modifications in gCOG are 1) differences in the

set of task operators, 2) the ability to use categorical tokens to allow for generic testing of multimodal reasoning, and 3) the ability to allow for arbitrarily long task trees to assess productive compositional generalization, in addition to distractor and systematic generalization (e.g., see Appendix Fig. 8). Importantly, the original COG task did not allow for tasks with more than a single conditional statement, e.g., a task tree of depth 3, making it difficiult to evaluate productive compositional generalization. The use of categorical stimulus tokens and instruction tokens generically tests the capacity of neural architectures to maintain, manipulate, and generalize novel compositions. Importantly, if models are unable to generalize using simple categorical encodings, then it is unlikely that these models will generalize when presented with the same task in a modality-specific domain. The total number of unique individual tasks (i.e., task trees of depth 1) is 8 operators $*$ 26 shapes $*$ 10 colors $=$ 2080 unique individual task commands, but can be straightforwardly extended with modifications to the configuration file. The number of total unique task structures explodes exponentially when task trees exceed depth 1 (e.g., 5,624,320,000 unique task structures for task trees of depth 3). We additionally provide functionality in the dataset that allows the choice to generate samples using either categorical task encodings, or task encodings with image pixels and natural language instructions. (All evaluations performed in this paper used categorical task encodings.)

The original COG dataset formulated tasks as directed acyclic graphs (DAGs). To simplify this representation (and to ensure a unique, topologically sorted solution of operators), we constrained tasks to binary trees. Unless otherwise stated, stimuli were mapped from a 10x10 spatial grid to a sequence of 100 binary tokens, where each token represented a specific location in the grid. If a location contained an object, the embedding was encoded as the specific attributes of that object. All task rule tokens were appended with an `EOS` statement, which was previously demonstrated to improve generalization (Csordás et al., 2022). (See Appendix for additional details on experimental design A.2, and how the task inputs were configured for model training and inference A.3.)

## 2.2 MODEL ARCHITECTURES

We evaluated performance of six encoder-only model architectures (Fig. 2). All models were trained to perform classification in a supervised manner. Outputs were projected to a vector with 138 elements, with each element in the vector representing a True/False boolean or a feature label (e.g., the color "Red", letter "a", or the spatial location (2, 1)). Models that included a Transformer component in the main figures used absolute positional encoding (Vaswani et al., 2017), though we also report results in the Appendix with Transformers that used relative positional encoding (Shaw et al., 2018; Huang et al., 2018) (Appendix Fig. 9 and Fig. 10). Importantly, there were no discernible differences between these choices of positional encoding. Finally, we report additional evaluations on deeper and larger SSTfmr models (i.e., BERT-style models) in Fig. 6 for all generalization splits. Those results demonstrate that improved distractor and systematic generalization performance can be achieved by scaling models (i.e., increasing encoder depth and attention heads), but not productive compositional generalization.

**RNNs and GRUs.** We trained both RNNs and GRUs with 512 units on `gCOG` (Fig. 2a). Task trees (i.e., instructions) were presented as a sequence of token embeddings, one for each node in the binary tree. The end of the rule sequence was marked with an `EOS` token. Stimuli were flattened from a $10 \times 10 \times D$ matrix to a $100 \times D$ matrix (where $D$ is the embedding dimension, and were presented simultaneously (i.e., not tokenized).

**Single stream Transformer (SSTfmr).** We trained a single stream Transformer, where the task instructions and stimuli were concatenated into a single matrix (with zero-padding), then passed through a shared Encoder block (Vaswani et al., 2017) (Fig. 2b). Thus, in the Transformer block, self-attention was applied on both rule and stimulus embeddings simultaneously. The output of the Transformer was then processed through a 2-layer MLP before projection to the output layer.

**Dual stream Transformer (DSTfmr).** We used a Transformer-based model to process task instructions and stimuli separately through modality-specific parallel Transformer blocks (Fig. 2c). Outputs from the Transformers were subsequently processed through a shared MLP.

**Transformers with Cross Attention (CrossAttn).** Like the DSTfmr, this Transformer-based model processed task instructions and stimuli separately through modality-specific Transformers. However, the outputs of the parallel Transformers were integrated through a cross-attention mechanism (Fig. 2d). Specifically, cross-attention was estimated by computing the query from the output of

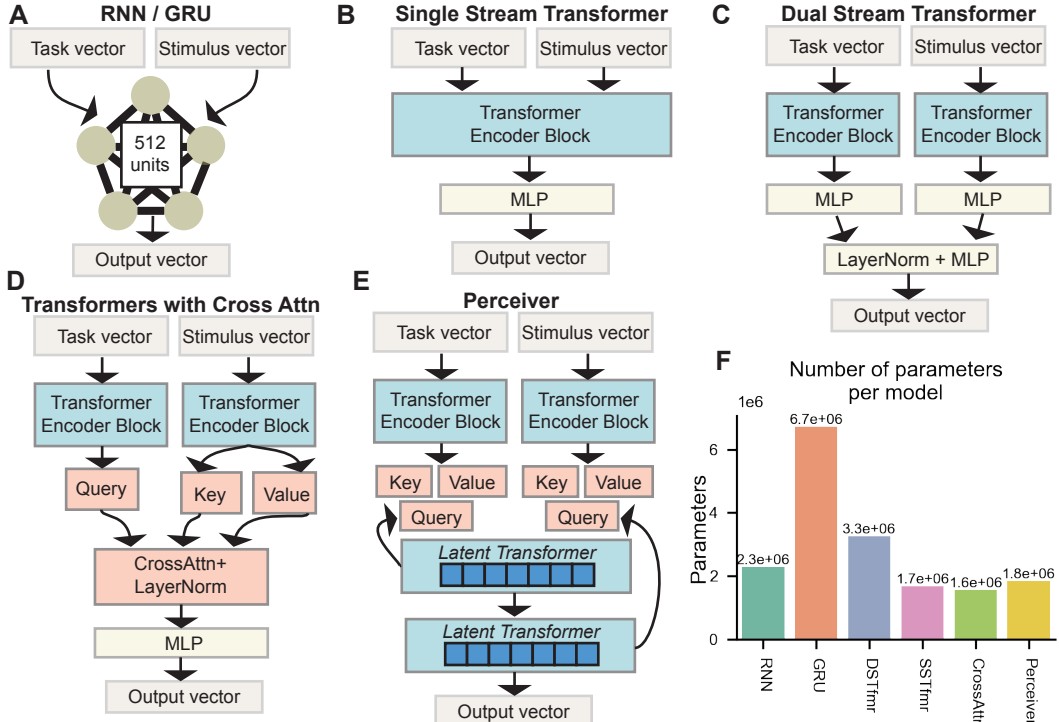

Figure 2: Model architectures. We evaluated generalization across six base neural network architectures. A) RNNs and GRUs with 512 hidden units. B) Single Stream Transformer (SSTfmr), which processes task rules and stimuli in a single Transformer, applying self-attention in the Transformer block. C) Dual Stream Transformer (DSTfmr). In contrast to the SSTfmr, two parallel Transformer blocks process rule and image tokens separately, and then process them together in a shared MLP. D) Transformers with cross-attention (CrossAttn). The outputs of two parallel Transformer blocks are processed with a cross-attention mechanism, where the output of the task rule Transformer produces a query, and then the stimulus Transformer block produces a key and value matrix. E) A Perceiver-like architecture, which integrates both task and stimulus output information in a latent Transformer through cross-attention (Jaegle et al., 2021). F) The number of parameters for each model.

the task rule Transformer, and the key and value matrix computed from the stimulus Transformer output. The cross-attention output was processed through a LayerNorm, and then an MLP.

**Perceiver-like architecture (Perceiver).** Finally, we included a Perceiver-like model (Jaegle et al., 2021), an architecture designed to generically process multimodal inputs (Fig. 2e). The Perceiver architecture contained a latent Transformer, which uses cross-attention to process information from input modalities. Specifically, the latent Transformer produces a query for each modality. The outputs of each modality-specific Transformer produced keys and values, which were subsequently processed through cross-attention with the latent Transformer. The latent Transformer also contained a standard self-attention Transformer block, followed by an MLP.

## 3 RESULTS

### 3.1 DISTRACTOR GENERALIZATION

**Experimental setup.** Distractor generalization evaluates the degree to which a model can generalize a task to a stimulus or environment with more distractors than it was trained on. For example, good distractor generalization requires that a model can correctly discern if a "red a" exists, independent of the number or configuration of distractors presented. We evaluated distractor generalization on an independent and identically distributed (IID) split and an OOD split. The IID split tests for generalization to stimuli with the same number of distractors that the model was trained on, but with a different configuration. The OOD split evaluates generalization to stimuli with more distractors than observed during training. Models were trained on individual task operators for simplicity (i.e., task trees of depth 1). Stimuli in the training set were randomly generated with a minimum of one

distractor and a maximum of five distractors. Models were trained on the same number of training steps (Appendix A.3). All models converged to greater than 94% training set accuracy (Fig. 3a-c).

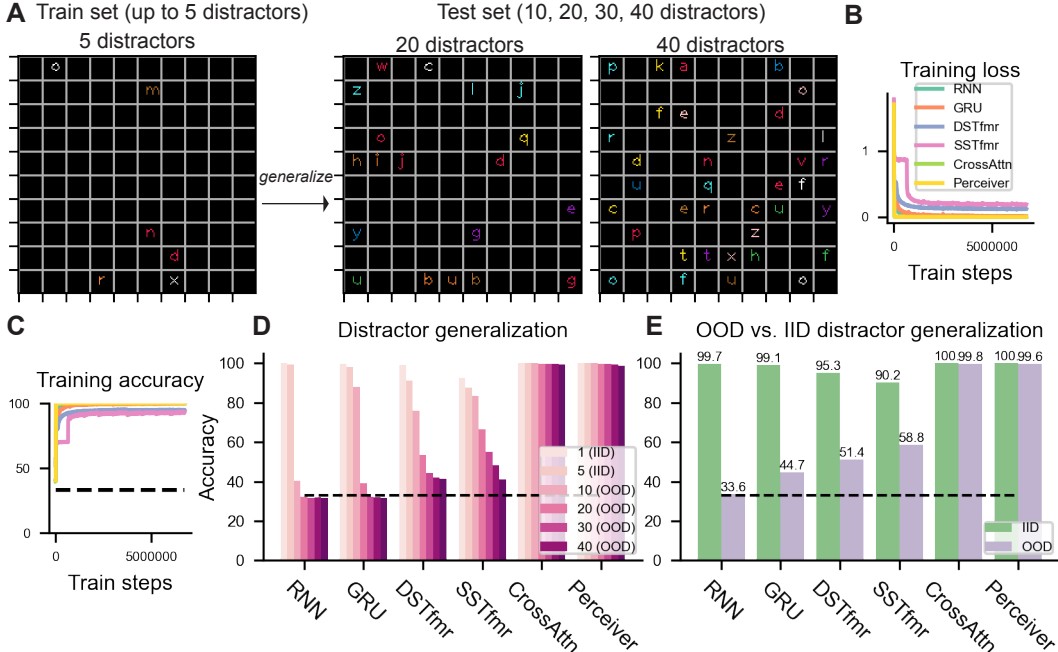

Figure 3: Distractor generalization. A) Experimental evaluation for distractor generalization. We trained models on individual task operators (e.g., "`Exist` red d") on stimuli that included 1 to 5 distractors, and then evaluated OOD generalization performance on stimuli with 10, 20, 30, and 40 distractors. B) Loss and C) accuracy trajectories during training for all models. All models converged to greater than 94% accuracy. D) Distractor generalization performance for each model. We assessed IID distractor generalization (novel stimuli, but with 1 or 5 distractors), and OOD distractor generalization (10, 20, 30, or 40 distractors). For most models, performance reduced as the number of distractors increased. E) We directly compared IID vs. OOD distractor generalization by averaging performance for IID and OOD splits. Models incorporating a cross-attention mechanism – CrossAttn and Perceiver – clearly exhibited the best performance.

**Generalization performance.** While all base models performed IID generalization well, only models that contained cross-attention mechanisms (CrossAttn and Perceiver models) exhibited excellent OOD distractor generalization (Fig. 3e). A related result was also reported in Qiu et al. (2021) using cross-modal self-attention. Though the GRU, SSTfmr, and DSTfmr models were able to perform some degree of OOD generalization (e.g., generalization on 10 distractors), performance was markedly reduced as the number of distractors increased from 10 to 20 (Fig. 3d).

## 3.2 SYSTEMATIC COMPOSITIONAL GENERALIZATION

The evaluation of Transformer models on systematic generalization problems has been of recent interest (Ontañón et al., 2022; Csordás et al., 2022; Hupkes et al., 2020; Keysers et al., 2020; Dziri et al., 2023). However, most evaluations have been limited to sequence-to-sequence tasks in a single modality (e.g., natural language). Here we extend prior work, and provide an evaluation of systematic generalization in the `gCOG` task using encoder-only architectures.

**Experimental setup.** Evaluating systematic compositional generalization requires a test set that is a novel recombination of previously seen tasks. In `gCOG`, this train/test split can manifest in several ways. The simplest split is to evaluate generalization on individual task operators (e.g., `Exist`) with objects (e.g., "red b") that it has not been paired with before. For example, if the model was trained on "`Exist` blue a" and "Get Location red b", it would have to systematically combine the notion of "red b" with the `Exist` operator – a configuration not seen in the training set (Fig. 4a).

Additionally, a more challenging test of systematicity is to evaluate generalization performance on more complex task tree structures. Prior question-answer benchmarks that evaluated systematic generalization typically were limited to assessing systematicity on individual task operations, rather

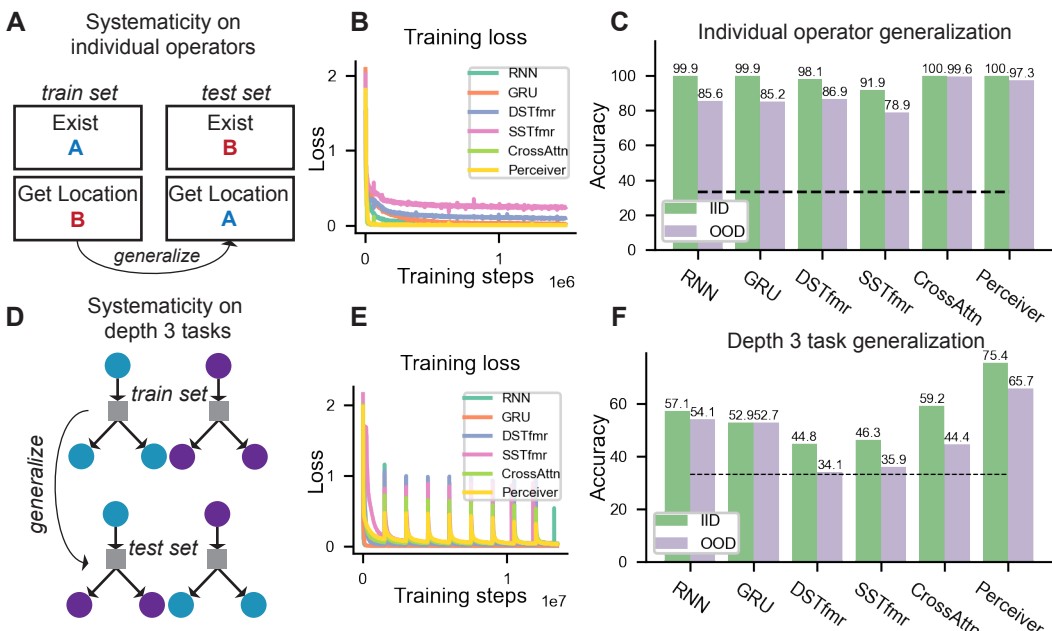

Figure 4: A) Systematicity on individual task operators, where specific objects (e.g., a blue "a") are trained on a subset of operators, and then tested on distinct set of operators. This evaluates if the model can generalize to new operator and object combinations. B) Training trajectories. C) CrossAttn and Perceiver-like models exhibit excellent systematicity generalization, while other models performed at reduced rates. D) Another test of systematicity is to train on task trees of depth 3, and then test on novel combinations of task trees of depth 3. E) Training trajectories. All models were able to efficiently learn this task variant. (Note that periodic spikes in the loss function are due to a resampling of the training dataset due to model checkpointing and/or disruption to a compute job.) F) While overall generalization performance is lower (even on IID generalization), cross-attention models still perform systematic compositional generalization well above chance.

than on task trees with greater dependencies (e.g., COG, SQOOP; Yang et al. (2018); Bahdanau et al. (2019; 2020). Here, we trained on a subset of task trees of depth 1 and 3, and then evaluated performance on an a novel combination of task structures of depth 3 (Fig. 4d). In this particular train/test split, we ensured that every individual task operator was previously trained on, but that a specific combination of task operators in a task tree was novel.

**Generalization performance.** On the systematicity test with individual task operators (Fig. 4a), all models converged on the training set with greater than 92% performance (Fig. 4b). All models exhibited reasonably good performance (>78%; chance=33.10%) on OOD systematic generalization (Fig. 4c). (Note that chance was determined as the average probability of each output classification given the distribution of the training set.) Across all models, models containing cross-attention mechanisms performed the highest on systematic generalization, with the CrossAttn and Perceiver architectures exhibiting the highest OOD generalization performance (>97%; Fig. 4c).

On systematic generalization on depth 3 tasks (Fig. 4d), IID generalization was markedly reduced across the board, despite convergence on the training set for all models (all models achieved >98% accuracy on the training set) (Fig. 4e,f). (Note, however, that increasing depth (encoder layers) to Transformers improves IID generalization on these splits; Fig. 6.) The Perceiver model outperformed all other models, exhibiting 75.4% IID systematic generalization, and 65.7% OOD generalization. The next best performing models had 59.2% IID generalization performance (CrossAttn model), and 54.1% OOD generalization performance (RNN). These results suggest that the Perceiver was best suited for systematic multimodal generalization, indicating its promise as a generic, amodal architecture for multimodal systematic generalization.

### 3.3 PRODUCTIVE COMPOSITIONAL GENERALIZATION

**Experimental setup.** Productive compositional generalization involves generalizing to tasks of greater complexity (e.g., a task tree of depth 3 to a task tree of depth 5). We evaluated productive

generalization in the context of gCOG. We trained all models on task trees of depth 1 and depth 3, and then evaluated generalization performance to task trees of depth 5 and depth 7 (Fig. 5a). (We also show in Appendix Fig. 11 how models trained only on task tress of depth 1 fail to generalize to task trees of depth 3, 5, and 7.)

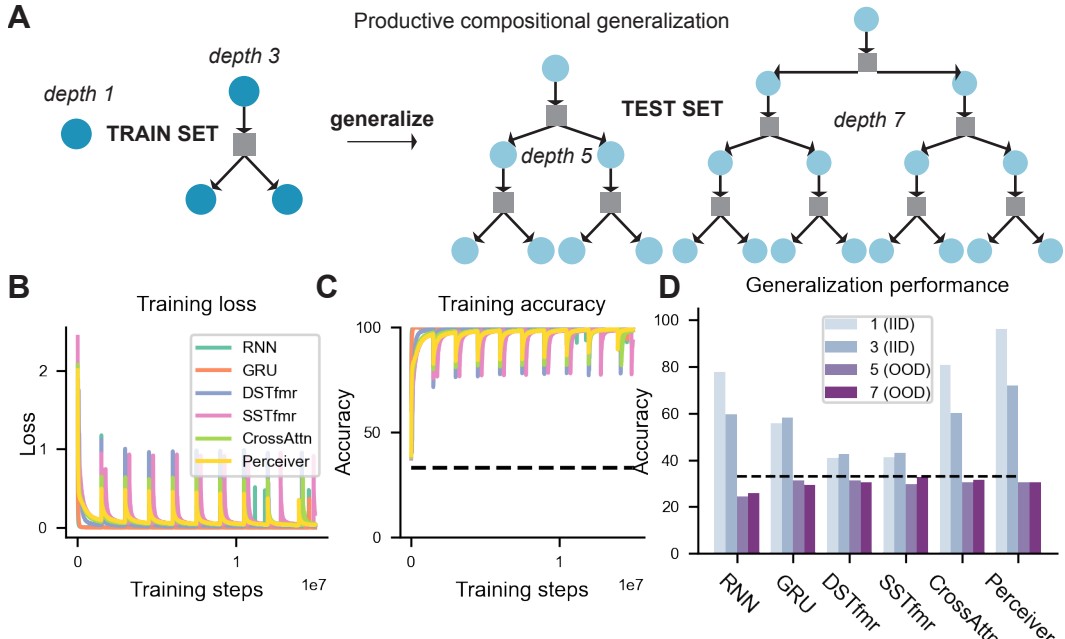

Figure 5: Productive compositional generalization performance. A) OOD productivity performance of all models to novel tasks of greater complexity (i.e., deeper task trees). We trained models on task trees of depth 1 and depth 3, and then tested generalization to task trees of depth 5 and 7. While the B) training loss and C) training accuracy converged for all models, D) all models failed to perform OOD productive compositional generalization to more complex task trees. (Note that periodic spikes in the loss function are due to a resampling of the training dataset due to model checkpointing and/or disruption to a compute job.)

**Generalization performance.** Though all models converged on the training dataset (Fig 5b,c), all models completely failed to OOD generalize to task trees of 5 and 7 (Fig. 5d) (performance was at or below chance for all OOD splits). Prior work has shown that Transformer-based models exhibit improved productivity generalization when using a relative positional encoding (Csordás et al., 2022; Ontañón et al., 2022). Though we used absolute positional encodings in our Transformer-based models in Fig. 5, we found that using relative positional encodings did not improve productive generalization performance on gCOG (Appendix Fig. 10). One possible explanation for the disparity between systematic and productive generalization in neural models is that systematicity requires the ability to exchange semantics (or tokens) from a known syntactic structure (e.g., a tree of certain depth). In contrast, productive generalization requires generalizing to an entirely new syntactic structure (e.g., a task tree of different size or depth). This requires understanding the syntax – how to piece together syntactic structures on-the-fly – requiring another level of abstraction. To our knowledge, there is no mechanism in modern Transformers that would enable this. Thus, productive compositional generalization remains a difficult capability for purely neural models to achieve.

### 3.4 IMPACT OF LAYER DEPTH AND ATTENTION HEADS ON GENERALIZATION

Our previous experiments evaluated the performance of base neural network models on gCOG. However, these "base" models did not assess the impact of model scale (e.g., depth) on performance. To complement those previous experiments, we evaluated the impact of scale (layer depth and attention heads) of a standard Transformer encoder model (e.g., BERT-style, and similar in base architecture to the SSTfmr; Devlin et al. (2019)) on generalization. We assessed the influence of encoder layers (1, 2, 3, and 4 layers), and the number of attention heads per encoder layer (1, 4, and 8 heads). We found that increasing encoder layers improves generalization across distractor (Fig. 6a,b) and systematic generalization (Fig. 6d,e,g,h). Increasing attention heads per layer also marginally im-

proved distractor and systematic generalization, but to a lesser extent than adding layers (Fig. 6c,f,i). Importantly, the largest model we tested (a BERT-small-sized model; 4 layers and 8 attention heads) demonstrated excellent systematic and distractor generalization. However, model scale failed to illustrate any improvement on productive generalization (Fig. 6j,k). These results demonstrate that increasing scale of standard Transformer architectures may be sufficient for distractor and systematic generalization with a well-designed dataset, but that productive generalization remains a significant challenge.

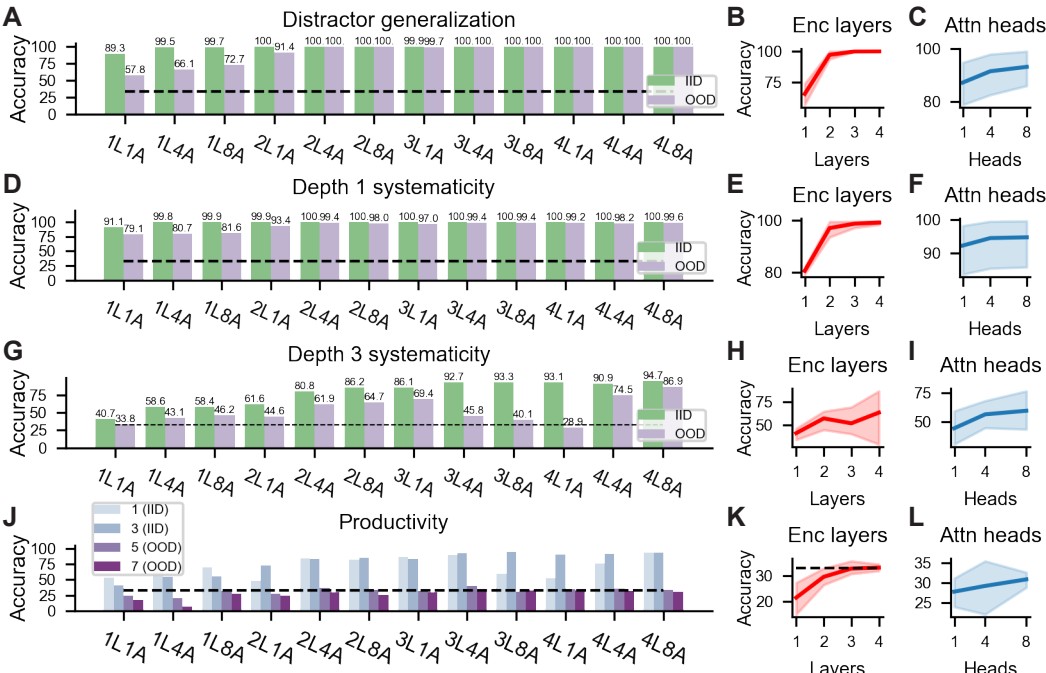

Figure 6: Evaluating generalization splits on BERT-like single-stream transformer models with varying layers (L) and attention heads (A). Overall, increasing layers and attention heads can improve generalization across distractor and systematic generalization, but not productive generalization. A) Evaluation on distractor generalization across all model parameters. B) The effect of adding additional encoder layers on distractor generalization performance (averaged across all attention head configurations). C) The effect of adding attention heads on distractor generalization performance (averaged across all layer depth configurations). D-F) Evaluation on systematicity for depth 1 tasks (generalization split in Fig. 4a). G-I) Evaluation on systematicity for depth 3 tasks (generalization split in Fig. 4d). J-L) Evaluation on productivity split (generalization split in Fig. 5a).

## 4 CONCLUSION

Identifying neural architectures that can robustly generalize OOD is a central goal in artificial intelligence. Compositional generalization benchmarks, which explicitly evaluate for generalization, provide a good testbed for measuring these capabilities. However, the most successful models for multimodal compositional generalization tend to be hybrid neuro-symbolic models rather than purely neural models (Bahdanau et al., 2020). While useful for some applications, current neuro-symbolic models require *a priori* knowledge of what rules and operations to include, making them difficult to train end-to-end, and limiting their broader use and overall scalability. In this study, we sought to evaluate how different architectural mechanisms in purely neural models influence OOD multimodal generalization. We introduced gCOG, which provides explicit OOD generalization splits for generic multimodal reasoning that can be extended in future studies. Our experimental results demonstrate that while current neural models fall short of exhibiting any productive compositional generalization, increasing layer depths and/or targeted cross attention mechanisms between multiple domains provide paths towards improving systematic and distractor OOD generalization on multimodal tasks. Thus, we hope this study inspires future work towards identifying the neural architectures capable of performing multimodal OOD generalization, with the goal of advancing the broader reasoning capacities of modern AI systems.

REPRODUCIBILITY STATEMENT

Code for this paper and dataset can be found at `https://github.com/IBM/gcog`. Additional details regarding the experimental design can be found in Appendix A.2. Additional details regarding the model architectures and training can be found in Appendix A.3. Additional details regarding representation analysis can be found in Appendix A.1.

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

# A APPENDIX

## A.1 REPRESENTATION ANALYSIS OF MODEL ARCHITECTURES

We performed representation analysis to identify the relationship between how a model's architecture constrains its internal representations, and in turn, its generalization behavior. Originally developed to analyze and interpret neuroscience data, representational similarity analysis (RSA) facilitates the interpretability of a model's internal activation patterns (Kriegeskorte et al., 2008; Klabunde et al., 2023). In brief, RSA summarizes the geometry of representations in a neural network layer (or layers) by measuring the similarities (or distances) between the activation vectors during the presentation of different inputs. This explicitly measures the relations between input samples, facilitating insight into how this neural network layer processes different samples. Moreover, transforming individual representations into a sample-by-sample matrix of representational similarities enables direct comparison between neural network layers both within and across networks, since the matrix dimensions can be made equal. (In contrast, layer activations across models or even within models are hard to compare, due to the lack of a 1-to-1 mapping of units.) In our specific case, we evaluated how different task features (stimulus input and target representaions) emerge in the penultimate layer of neural network models. This was done by computing the similarities of stimuli (the vectorized stimulus encoding) and target responses (one-hot codes for each target response), and then computing the distance of these similarity matrices to the representational similarity matrices of the model's penultimate layer.

In our analysis, we sought to understand why models with different architectures (e.g., the presence of a cross-attention mechanism) better performed OOD distractor generalization. We focused our analysis to the distractor generalization benchmark, since we observed the strongest discrepancy in OOD generalization performance across models. To evaluate what task information was retained within the representations of each model during OOD distractor generalization, we randomly sampled 800 task samples with 40 to 50 distractors and quantified the vectorized cosine similarities between every pair of task stimuli (in the input space) (Fig. 7a). (Note, however, that other distance metrics can also be used Klabunde et al. (2023).) This resulted in a stimulus similarity matrix, from which we could directly compare a model's internal representational similarities using the same 800 samples. Next, we measured each model's representational similarity matrix to the same set of stimuli using activations in the models' penultimate layer (Fig. 7a). To measure how much stimulus information the neural network's layer retained, we computed the L2 norm between matrices after aligning the two matrices through the orthogonal Procrustes transform. (The same procedure could be applied to measure the representational alignment to target resposne information (Fig. 7c).) The orthogonal Procrustes transform between the representational similarity matrix, $X$, and stimulus (or target) matrix $Y$, was computed by solving for the orthogonal transformation

$$Q^* = \arg \min_{Q \in O(D)} ||XQ - Y||_F$$

where $Q^*$ is the best orthogonal transformation to align $X$ and $Y$, $O(D)$ is the group of orthogonal transformations, and $||\cdot||_F$ denotes the Frobenius norm. Then, to compute the alignment $A$ between matrices $X$ and $Y$, we compute

$$A = ||XQ^* - Y||_F$$

We used this distance metric as it has been previously shown to be a proper distance metric (i.e., it obeys the triangle inequality; Klabunde et al. (2023). However, this measure alone does not control for potential biases in representational distances due to architecture. Thus, to control for the potential confound that some model architectures have different innate distances, we computed the relative alignment by computing the difference of $A$ before and after training the model. Importantly, relative alignment can be negative, since it provides a metric of how the representations change relative to the randomly initialized model architecture. We use this measure of relative alignment to measure how much stimulus or target response information each model architecture retained in Figure 7. We found that the best performing models (CrossAttn and Perceiver models) on OOD distractor generalization, had the highest alignment to the stimuli, while the lowest performing models (e.g., RNNs and GRUs), were least aligned to stimulus information (Fig. 7b). We found a similar trend when estimating the model's alignment to the correct response (i.e., target) (Fig. 7c). Thus, the best OOD distractor generalization models were able to retain both stimulus and response information in the penultimate internal representations.

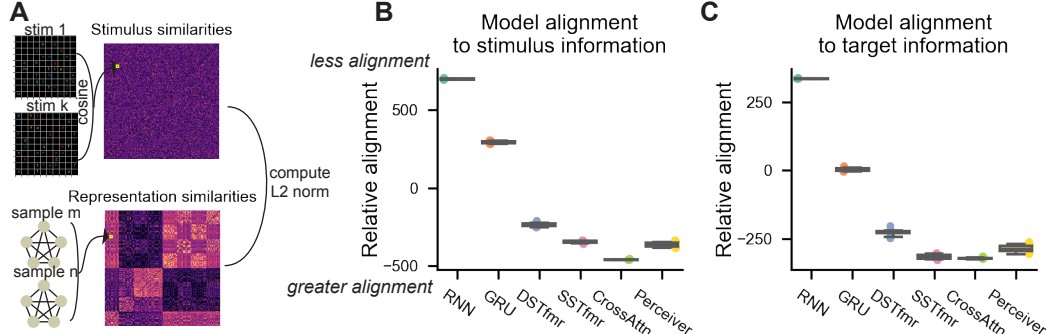

Figure 7: Representation analysis across model architectures. A) To measure task information in each model's internal representations, we measured the representational similarity matrices of the stimulus (or target) samples, and compared those to the models' representational similarity matrices during those same samples. Retainment of stimulus (or target) information in the model's representations could then be measured as the L2 norm between model and the task stimulus (or target) matrices. B) Model alignment to stimulus information. Models with cross-attention mechanisms best retained stimulus information in the penultimate layer. C) Model alignment to target (response) information.

## A.2 ADDITIONAL DETAILS ON TASK DESIGN

`gCOG` was modified from the original COG task to enable the measurement of various OOD benchmarks (Yang et al., 2018). While we employed some similar task operations (e.g., the `Exist` operator), we also implemented a number of new task operators that were not in the original dataset. We also lifted the constraint of maximally including only a single conditional (i.e., an `IF-THEN-ELSE` statement, originally referred to as a "Switch" clause in COG), allowing us to flexibly evaluate productive compositional generalization. A few example queries from the original COG task include: "What is the color of the latest triangle? Point to the latest red object. If a square exists, then point to the current x, otherwise point to the last b."

Construction of task samples was algorithmically consistent with how samples were constructed in COG and CLEVR (Johnson et al., 2017). Construction of a task sample (i.e., task instructions, stimuli, and target) began by first constructing the task instruction, and then determining the specific task path (i.e., which operators will be encountered in the task tree). Depending on what the target response is, a backward pass is taken through the task path (i.e., bottom-up). Objects are then incrementally added to the stimulus to satisfy all task nodes from the bottom-up. This ensures that the stimulus will necessarily satisfy the chosen task path while guaranteeing a unique solution for each task node. More specifically, there will always be a unique object or object feature to satisfy a task operator. (If the task operator is to *Get the Color of the "a"*, the task algorithm guarantees that there will only be a single "a" in the image. If the task operator is *Does the red "a" exist?*, there is guaranteed to be a maximum of one red "a".) Additional algorithmic details on task construction can be found in Yang et al. (2018) and Johnson et al. (2017), and the task code will be publicly released. We will additionally release specific benchmarks that were used in this study.

Note that in the IID split for each test, it is theoretically possible that the model will be tested on a sample that it was trained on due to random sampling, though this is highly unlikely: the probability that two identical stimuli are presented is less than $10^{-14}$ due to the total number of distractor combinations.

### A.2.1 TASK OPERATORS

**Exist.** The `Exist` operator is paired with an object (i.e., color and shape feature combination), and asks whether that object exists in the stimulus. The correct response returns a boolean (`True` or `False`). Example: *Does the red "a" exist?*

**GetColor.** The `GetColor` operator is paired with a shape feature, and asks to return the color of that shape. The correct response is a color feature (1 out of 10 possible color features). Example: *Get the color of the "a"*. (In the task construction, there is guaranteed to be a unique solution (i.e., one "a") in the image.

**GetShape.** The `GetShape` operator is paired with a color feature, and asks to return the shape of the object with that color. The correct response is a shape feature (1 out of 26 possible shape attributes). Example: *Get the shape of the red object*. (In the example, there will always be a single red object.)

**GetLocation.** The `GetLocation` operator is paired with an object, and asks to return its location. The correct response is the (x, y) coordinates of that object (1 out of 100 possible locations in a 10 by 10 grid). Example: *Get the location of the red "a"*.

**SumEven.** The `SumEven` operator is paired with an object, and asks whether the sum of its x and y coordinate location is even. The correct response is a `True`/`False` boolean. Example: *Is the sum of the x and y coordinate values of the red "a" even?*

**SumOdd.** The `SumOdd` operator is paired with an object, and asks whether the sum of its x and y coordinate location is odd. The correct response is a `True`/`False` boolean. Example: *Is the sum of the x and y coordinate values of the red "a" odd?*

**ProductEven.** The `ProductEven` operator is paired with an object, and asks whether the product of its x and y coordinate location is even. The correct response is a `True`/`False` boolean. Example: *Is the product of the x and y coordinate values of the red "a" even?*

**ProductOdd.** The `ProductOdd` operator is paired with an object, and asks whether the product of its x and y coordinate location is odd. The correct response is a `True`/`False` boolean. Example: *Is the product of the x and y coordinate values of the red "a" odd?*

Note that some of these operators can be used as terminals, i.e., task tree leaves with no children. This includes all task operators that do not return a boolean response (`GetColor`, `GetShape`, and `GetLocation`). This is because a boolean response is required to be input to an `IF-THEN-ELSE` clause.

### A.3 ADDITIONAL DETAILS ON MODEL ARCHITECTURES AND TRAINING

Code associated with model architectures can be found here: `https://github.com/IBM/gcog`. Models were constructed using PyTorch version 2.0.0+cu118. All models could be trained in under three days on an NVIDIA K80 GPU, and were trained on IBM's Cognitive Compute Cluster.

For a specific evaluation benchmark (e.g., distractor generalization), all models were trained on exactly the same number of samples (and training steps). This made it possible to fairly evaluate and compare performance of different models. For distractor generalization (Fig. 3), all models were trained on 53,980,000 samples. For systematic generalization on individual operators (Fig. 4a), all models were trained on 47,980,000 samples. For systematic generalization on depth 3 tasks (Fig. 4d), all models were trained on 53,980,000 samples. For productive generalization (training on tasks with depth 1 and 3; Fig. 5), all models were trained on 59,980,000 samples.

All models were trained using the AdamW optimizer with a learning rate of 0.0001, and the loss was computed as the Cross Entropy between the target class and output vector.

#### A.3.1 MODEL ARCHITECTURES

The inputs to the models comprised of rule tokens (i.e., task instructions) and stimulus. For Transformer-based models (SSTfmr, DSTfmr, CrossAttn, and Perceiver) the 10x10 stimulus grid was presented as a sequence of 100 tokens with absolute positional encoding. Each stimulus token had three features (i.e., embedding dimensions) associated with it: color, shape, and a separate dimension indicating if the token was an `EOS` token. For RNN and GRU models, stimuli were flattened into an array ($100 \times 3$). Rule tokens were embedded into a feature space containing an dimension for the task operator (e.g., `Exist`, `GetColor`, etc.), and separate dimensions for the operator's object(s) features (e.g., color, shape, etc.). All models encoded rule tokens as a sequence.

**RNN.** Inputs to the RNN were a sequence of rule tokens and a flattened stimulus vector, which were projected to 512 hidden units. Hidden unit activity, $h_t$ was determined by the equation

$$h_t = tanh((x_{r,t}W_{r,h}^T + b_{r,h}) + (x_{s,t}W_{s,h}^T + b_{s,h}) + h_{t-1}W_{hh}^T + b_{hh}$$

Example of task trees of varying depth

Figure 8: Example task trees of depth 1, 3, 5, and 7. In the current design, there are 8 task operators that can be paired with up to 260 unique objects (26 letters * 10 colors). However, some operators can only be used as tree leaves (i.e., they have no children in the task tree). These include `GetColor`, `GetShape`, and `GetLocation`. This is because if a task operator has a child which is an if-else-then clause, the task operator is required to return a boolean.

where $x_{r,t}$, $W_{r,h}^T$, and $b_{r,h}$ represented the rule inputs, weights, and biases respectively, and $x_{s,t}$, $W_{s,h}^T$, and $b_{s,h}$ the stimulus inputs, weights, and biases. The stimulus vector was presented every time a rule token was processed. This avoided the need for the RNN to "remember" the stimulus. Hidden units were initialized as a vector of zeros prior to each trial. The RNN hidden activation patterns were processed through a LayerNorm after every iteration, and then subsequently projected to the output layer (with 138 units) for classification. A Softmax was applied prior to supervised training of the outputs. Model training was performed with the AdamW optimizer at a learning rate of 0.0001 (Loshchilov & Hutter, 2019), and the loss was computed as the Cross Entropy between the target class and the output vector.

**GRU.** Architecturally, the GRU was structured identically to the RNN. Input and outputs were formatted and projected identically, and the placement of LayerNorms remained the same. The primary distinction was the generation of the hidden unit activity $h_t$, which was determined by the equation

$$h_t = (1 - z_t) \odot n_t + z_t \odot h_{t-1}$$

where $n_t$, $z_t$, and $r_t$ correspond to the new, update, and reset gates, respectively, and are defined by the equations

$$n_t = tanh(W_{in}x_t + b_{in} + r_t \odot (W_{hn}h_{t-1} + b_{hn}))$$

$$z_t = \sigma(W_{iz}x_t + b_{iz} + W_{hz}h_{t-1} + b_{hz}))$$

$$r_t = \sigma(W_{ir}x_t + b_{ir} + W_{hr}h_{t-1} + b_{hr}))$$

where $\sigma$ is the sigmoid function, and $\odot$ is the Hadamard product. As in the case of the RNN, the GRU model was trained with the AdamW optimizer with a learning rate of 0.0001, and the loss was computed as the Cross Entropy between the target class and the output vector.

**SSTfmr.** Rule inputs the the SSTfmr were identical to the RNN, i.e., a sequence of rule tokens. Stimulus input was represented as a sequence of tokens. Specifically, each location of the stimulus (i.e., each (x,y) coordinate) corresponded to a single token. Since there were 10 rows and 10 columns, there were a total of 100 tokens. We zero-padded concatenated both the rule tokens and the stimulus tokens into a single context window with zero-padding. Thus, the total number of input tokens corresponded to the sum of rule and stimulus tokens. The architecture of the SSTfmr followed a standard bidirectional Encoder-only Transformer, and is architecturally similar to BERT (Devlin et al., 2019) (without masking). Since multimodal inputs were concatenated before being fed into the SSTfmr, self-attention was technically cross-modal. Note, however, that cross-modal self-attention is distinct from cross-attention, since self-attention always produces a square (quadratic) attention matrix, while cross-attention can produce a square matrix, depending on the number of tokens in the query matrix and the number of tokens in the key and value matrices. Input tokens were linearly embedded into a vector size of 256. For simplicity, we only included a single Transformer layer, and each layer only included a single attention head, though we do a more extensive evaluation of this type of architecture in Figure 6. We evaluated models with both absolute positional encoding (following Vaswani et al. (2017)) and relative positional encoding (following Shaw et al. (2018)). The position-wise MLP portion of the Transformer block was a 2-layer MLP with 512 units in each layer. The output of the Transformer block was subsequently processed through a 3-layer feedforward MLP (512, 1024, and 512 units per layer). We applied a LayerNorm prior to projecting the activity to the output layer for classification. Model training was performed with the AdamW optimizer at a learning rate of 0.0001, and the loss was computed as the Cross Entropy between the target class and the output vector.

**DSTfmr.** The DSTfmr was architecturally similar to the SSTfmr, except rule and stimulus tokens were processed independently in separate (and in parallel) encoder-only Transformer blocks, followed by separate 3-layer MLPs. (No zero-padded concatenation was applied to the rule and stimulus tokens.) The output of the parallel MLPs (Fig. 2c) were then summed together, processed through a LayerNorm, and then passed through a 3-layer MLP (512, 1024, and 512 units per layer). Model training was performed with the AdamW optimizer at a learning rate of 0.0001, and the loss was computed as the Cross Entropy between the target class and the output vector.

**CrossAttn.** Inputs to the CrossAttn model were processed identically to the DSTfmr model, i.e., in dual stream encoder-only Transformer blocks. Cross-attention was then computed from the outputs of the two Transformer blocks. Specifically, the query was computed from the output of the rule Transformer block, and the keys and values were computed from the output of the stimulus Transformer block (Fig. 2d). The cross-attention output was then processed through a LayerNorm (with a skip connection from the rule Transformer block output), and a 2-layer MLP (both with 512 units). After a final LayerNorm, the activations were linearly projected to the classification layer. Model training was performed with the AdamW optimizer at a learning rate of 0.0001, and the loss was computed as the Cross Entropy between the target class and the output vector.

**Perceiver.** Inputs to the Perceiver-like model were processed identically to the DSTfmr model, i.e., in dual stream encoder-only Transformer blocks. Outputs of the dual-stream Transformer blocks were processed through separate cross-attention mechanisms with the latent Transformer (Fig. 2e). The processing of inputs to the latent Transformer followed the structure of the Perceiver model (Jaegle et al., 2021). In our specific case, the latent Transformer contained 256 latent units, and was initialized to zero at the start of every trial. We computed cross-attention between the latent Transformer and the output of the rule Transformer first, and then between the latent Transformer and the output of the stimulus Transformer. Cross-attention involved computing the query from the latent units, and the keys and values from the modality-specific Transformers. LayerNorms and

residual connections were computed after every cross-attention computation, followed by a self-attention computation. After self-attention was computed after integrating information, the latent activations were projected to the output layer for classification.

**BERT-like SSTfmr**. To assess the role of Transformer depth and attention head number in generalization splits. we performed additional experiments on a range of BERT-like SSTfmr models (Fig. 6). This included systematically varying number of Transformer layers (1, 2, 3, and 4 layers), and attention heads per layer (1, 4, and 8 attention heads). The model with 4 encoder layers and 8 attention heads was architecturally identical to BERT-small, with the exception that the embedding dimensionality was limited to 256 (rather than 512). Moreover, unlike the SSTfmr used in the main manuscript (described above), we removed the additional MLP that was included on top of the Transformer's encoder layer. The outputs of the final encoder layer were projected to the ouput layer for classification (using just a linear layer with a Softmax). For the experiments included in Fig. 6, all models used relative positional encoding (Shaw et al., 2018). BERT-like SSTfmr models were trained on a total of 11,980,000 samples before test set evaluation.

## A.4 SUPPLEMENTARY RESULTS

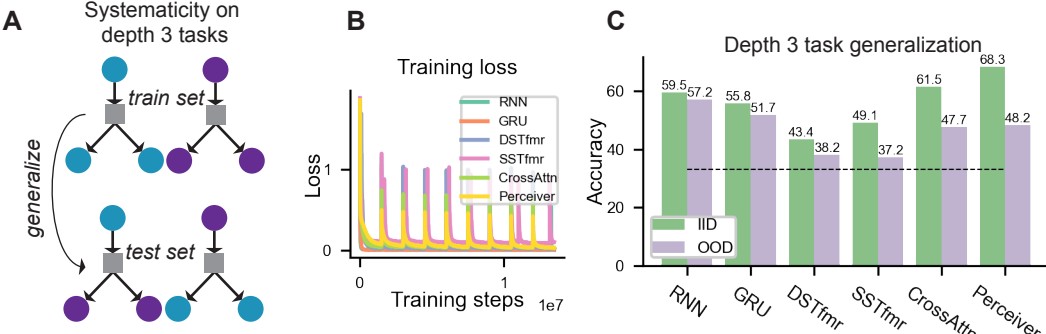

Figure 9: Systematic compositional generalization performance using relative positional encoding (Shaw et al., 2018; Huang et al., 2018) A) Systematicity evaluation on task trees of depth 3. B) Training trajectories. C) While we observe overall similar generalization performance patterns as observed using absolute positional encoding, we see some reduction in performance in the cross-attention models using relative positional encoding.

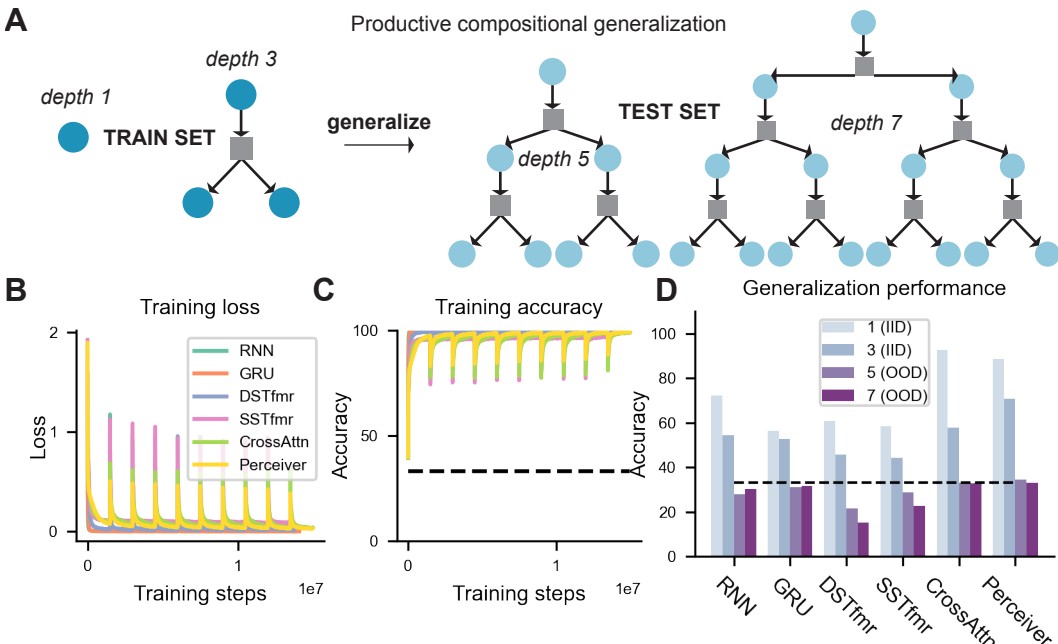

Figure 10: Productive compositional generalization performance using relative positional encoding (Shaw et al., 2018; Huang et al., 2018) A) OOD productivity performance of all models to novel tasks of greater complexity (i.e., deeper task trees). We trained models on task trees of depth 1 and depth 3, and then tested generalization to task tress of depth 5 and 7. While the B) training loss and C) training accuracy converged for all models, D) all models failed to perform OOD productive compositional generalization to more complex task trees. The results with relative positional encoding are overall similar to the results with absolute positional encoding (Fig. 5).

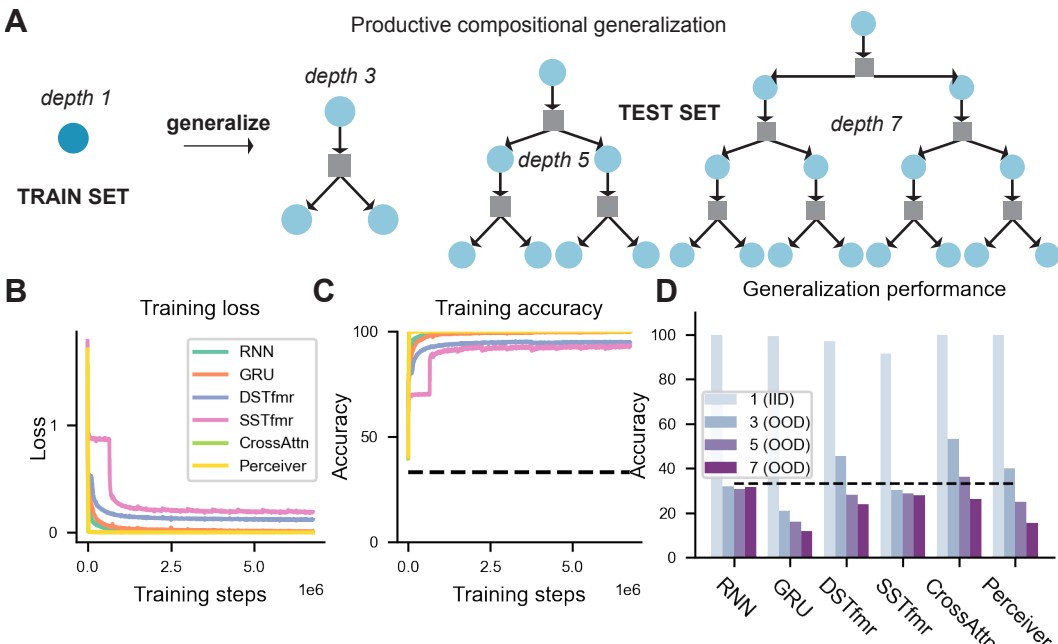

Figure 11: Productive compositional generalization performance, training on task trees of depth 1 (absolute positional encoding), and testing on task trees of depth 3, 5, and 7. A) In contrast to Figures 5 and 10, we exclusively train on task trees of depth 1, and then assess generalization to task tress of depth 3, 5, and 7 (i.e., deeper task trees). Evaluation on task trees of 3, 5, and 7 are therefore OOD. While the B) training loss and C) training accuracy converged for all models ($> 94\%$), D) all models performed poorly on OOD productive compositional generalization.

