# OpenReview forum: "On the generalization capacity of neural networks during generic multimodal reasoning"
_ICLR.cc/2024/Conference — ICLR 2024 poster_

### Official Review · Reviewer_skmj · 2023-10-30

**Soundness:** 3 good
**Presentation:** 3 good
**Contribution:** 3 good
**Rating:** 6
**Confidence:** 4

**Summary:**

The paper studies multi-modal generalization in neural networks such as transformer-based models and recurrent networks. To do so, the authors propose Genertic COG, a modular benchmark with multi-modal splits to test for 3 types of generalization: 1) distractor (generalization to different noise distribution), 2) systemic compositional (generalization to new permutation of task structures) and 3) productive compositional (generalization to tasks of greater complexity) generalization. Experiments conducted by the authors showed that while cross-attention based transformers (e.g. CrossAttn and Perceiver) outperform other models and perform well on distractor and systemic compositional generalization, they fail at productive generalization when the depth of the task tree goes to out-of-distribution (>3). Representational analysis is done to show that cross-attention based transformers (e.g. CrossAttn and Perceiver) superior performance on distractor generalization might be due to their ability to better retain task-relevant (e.g. stimulus and response) information at the penultimate layer.

**Strengths:**

+The paper studies a timely and critical question about the generalization capability of multimodal transformer-based models

+The proposed benchmark dataset uncovers a limitation of current multimodal transformer-based models: productive generalization which can facilitate the development of more generalizable transformers/LLMs.

+The paper is generally well-written and easy to follow

**Weaknesses:**

-While the paper’s studies show that certain designs (e.g. cross-attention) seem to confer multi-modal generalization, there are still some key questions that can be more thoroughly studied to uncover the reasons why this is the case.

-Similarly, important discussions such as why the (cross-attention) transformers might fail at productive generalization is lacking.

**Questions:**

What is the key architectural difference between dual stream transformer and transformers with cross attn that can explain their generalization performance? Is it only the lack of a cross attention between the different modalities?

Possible typo:
“Finally, we included a Perceiver-like model (Jaegle et al., 2021), an architecture designed to generically process multimodal inputs (Fig. 2f).”:  (Fig. 2f) > (Fig. 2e).


==Post-Rebuttal==
I appreciate the authors' response and decided to keep my score.

---

> ### Author Response · Authors · 2023-11-21
> **Response to Reviewer skmj (1/N)**
>
> We thank the Reviewer for providing feedback on our manuscript, and the overall positive assessment. Below we address the weaknesses and specific questions the Reviewer raised.
>
>  **Weaknesses**
>
> *While the paper’s studies show that certain designs (e.g. cross-attention) seem to confer multi-modal generalization, there are still some key questions that can be more thoroughly studied to uncover the reasons why this is the case.*
>
> In response to the Reviewer’s concerns (and the related comment by Reviewer a4Su), we have now performed additional experiments that focus on how model scale and complexity can influence multimodal generalization. While the original manuscript was focused on understanding how a class of base neural architectures would fare on multimodal generalization, we agree that it is important to understand how choice of hyperparameters, such as number of attention heads and layer depth, can influence generalization.
>
> As such, we have performed a systematic experiment on a standard encoder-only transformer (with the same architecture as BERT; Devlin et al., 2019). We manipulated the number of layers (1, 2, 3, 4 layers), and number of attention heads (1, 4, 8 heads) and assessed the corresponding generalization performance across all splits. (Note that 4 transformer encoder layers and 8 attention heads match the BERT-small architecture.) Indeed, we found that increasing encoder depth significantly improved distractor and systematic generalization. Increasing attention heads also improved distractor and systematic generalization, though to a lesser extent. Nevertheless, neither of these modifications influenced productive generalization. Our conclusion from this is that some tasks (e.g., systematic generalization) require more abstractions; a single layer of attention that handles multimodal inputs is insufficient. Cross-attention mechanisms offer a more targeted and efficient solution (I.e., fewer number of parameters and faster to train) that explicitly integrate visual stimuli (keys and values) with the instructions (queries). However, simply adding encoder/attention layers and parameters can suffice for some types of generalization.
>
> We have now included discussion of some of these experiments and insights into the manuscript, with a few key revisions copied below. The new results are depicted in Figure 8 (Appendix). Below is the caption for Figure 8; please see the updated PDF submission for the actual figure: “Figure 8: Evaluating generalization splits on BERT-like single-stream transformer models with varying layers and attention heads. We manipulate a generic encoder-only transformer based on the BERT architecture, evaluating the influence of the number of encoder layers (1, 2, 3, and 4 layers), and the number of attention heads per encoder layer (1, 4, and 8 heads). Overall, increasing layers improves generalization across distractor and systematic generalization, but not productive generalization. Increasing attention heads also marginally improves distractor and systematic generalization, but to a lesser extent than adding depth. A) Evaluation on distractor generalization across all model parameters. B) The effect of adding additional encoder layers on distractor generalization performance (averaged across all attention head configurations). C) The effect of adding attention heads on distractor generalization performance (averaged across all layer depth configurations. D-F) Evaluation on systematicity for depth 1 tasks (identical to generalization split in Fig. 4a). G-I) Evaluation on systematicity for depth 3 tasks (identical to generalization split in Fig. 4d). J-L) Evaluation on productivity split (identical to generalization split in Fig. 5a).”
>
> Updates to Contributions Section (1.2): “2. A comprehensive evaluation of commonly-used base neural models (RNNs, GRUs, Transformers, Perceivers) on distractor, systematic, and productive generalization splits. We find that for distractor and systematic generalization, including a cross-attention mechanism across input modalities is important. However, all models fail on the productivity split. In addition, we include experiments demonstrating the impact of transformer depth and attention heads on all generalization splits in an encoder-only Transformer model."

---

> ### Author Response · Authors · 2023-11-21
> **Response to Reviewer skmj (N/N)**
>
> *Similarly, important discussions such as why the (cross-attention) transformers might fail at productive generalization is lacking.*
>
> This is a challenging question to tackle. Our ongoing hypothesis is that productive generalization is a fundamentally distinct type of generalization relative to systematic compositional generalization. We have now included a brief discussion in the Results section of Productive Compositional Generalization (Section 3.3) that addresses these challenges: “One possible explanation for the disparity between systematic and productive generalization in neural models is that systematicity requires the ability to exchange semantics (or tokens) from a known syntactic structure (e.g., a tree of certain depth). In contrast, productive generalization requires generalizing to an entirely new syntactic structure (e.g., a task tree of different size or depth). This requires understanding the syntax -- how to piece together syntactic structures on-the-fly -- requiring another level of abstraction. To our knowledge, there is nothing in the current set of mechanisms in the Transformer that would enable this. Thus, productive compositional generalization remains a difficult capability for purely neural models to achieve.”
>
> **Questions**
>
> *What is the key architectural difference between dual stream transformer and transformers with cross attn that can explain their generalization performance? Is it only the lack of a cross attention between the different modalities?*
>
> The short answer is yes. When comparing the Dual Stream Transformer with the models with cross attention, indeed, the only distinction is the lack of an attention mechanism to explicitly integrate outputs from the two input streams.
>
> The longer answer, which became clear after performing the new experiments (that scaled up to the BERT-small architecture), is that at minimum, a second attention layer is required to systematically abstract token information from each of the inputs. While applying a large self-attention matrix twice to simultaneously presented visual and language instructions is computationally inefficient (our cross-attention models show that it is unnecessary), it provides the necessary base structure to allow for good systematic generalization.
>
> *Possible typo: “Finally, we included a Perceiver-like model (Jaegle et al., 2021), an architecture designed to generically process multimodal inputs (Fig. 2f).”: (Fig. 2f) > (Fig. 2e).*
>
> We thank the Reviewer for spotting this error. The manuscript has now been updated.

---

### Official Review · Reviewer_a4Su · 2023-11-07

**Soundness:** 1 poor
**Presentation:** 2 fair
**Contribution:** 2 fair
**Rating:** 3
**Confidence:** 3

**Summary:**

This paper proposes a new benchmark for assessing various forms of generalization in a multimodal setting named gCOG. The dataset includes several different splits intended to measure different aspects of generalization. The paper also compares several different model architectures on the dataset.

**Strengths:**

* The paper introduces a new dataset, gCOG. While the dataset is conceptually similar to those from prior work, such as gSCAN, it supports different types of contexts and instruction types, including more compositional instructions. I'm aware of some prior work (e.g. [1], [2]) that studied compositional generalization in natural language tasks and found that gains on one synthetic task did not always transfer to other tasks, so increasing the diversity of such benchmarks for assessing compositional generalization and related challenges in the multimodal setting could be a potentially valuable contribution.

[1] https://arxiv.org/abs/2007.08970
[2] https://aclanthology.org/2021.acl-long.75/

**Weaknesses:**

* I'm concerned about the strength of the baselines used in the paper (see my related questions below). While the primary contribution of the paper is the dataset, it is also important to establish strong baselines for this new dataset and to ensure that the conclusions from the empirical results are valid. The appendix states that only a *single Transformer layer* with a *single attention head* was used. This is almost certainly not an optimal depth and number of attention heads. Relatedly, it looks like some models are potentially underfit, according to the figures. With >5M training examples and a relatively simple input space, I would have expected a reasonably sized Transformer model to achieve low training loss and reasonable IID generalization. If these models could have been applied to similar tasks such as gSCAN (even using symbolic tokens to represent the scene context), where they could be compared with comparable baselines from prior work, this would have helped establish that these are indeed reasonably strong baselines that have been well tuned.
* The qualitative difference between gCOG and datasets from prior work such as gSCAN was not very clearly described. For example, one of the key claims seemed to be gCOG "employs generic feature sets that are not tied to any specific modality". However, it seems like it is a useful property for a multimodal dataset to have a clear relation to real-world multimodal tasks. Indeed, the authors provide interpretations of their tasks in the form of natural language instructions and visual scenes (e.g. in Figure 1), and these are very useful for understanding the task. Representing this dataset using familiar modalities (e.g. vision, natural language) could enable future work to study different research questions, e.g. the impact of pre-training. The ability to alternatively represent the task input as a sequence of tokens is also reasonable for studying certain research questions, but this also seems possible for datasets from prior work. For example, I understand that gSCAN includes both symbolic descriptions as well as visual renderings. Anyways, I think clarifying the motivation for this dataset (e.g. increasing diversity of available benchmarks, focusing on different generalization challenges, etc.) separately from how inputs are represented for the experiments in this paper (e.g. token sequence vs. images and natural language) would be useful.
* Some of the main empirical conclusions (e.g. that generalization to greater "depth" is challenging for models such as Transformers) are generally known from prior work.

nits:
* Introduction paragraph 1 - "on a carefully controlled generic multimodal reasoning tasks" -> "on carefully..." or "...task"
* Appendix A.2.1 - Maybe reference Tables 8 and 9 where you discuss different positional embeddings.
* Consider discussing [3] in related work. [3] demonstrated the importance of cross-modal attention for gSCAN, and similarly studied the relative difficulty of various aspects of generalization, including distractors.

[3] https://aclanthology.org/2021.emnlp-main.166/

**Questions:**

* Why not try more layers and attention heads, e.g. following a standard hyperparameter setting for model size such as those of BERT-Base? Or even BERT-Small?
* In Figure 2 (F) why does the single-stream Transformer have almost double the parameters of the double stream Transformer? For the other Transformers, do the encoder blocks used for the task vector and stimulus vector share parameters?
* What optimizer and hyperparameters (e.g. learning rate) were used for training? How were these chosen? I didn't see these details in Appendix A.2.
* Position embeddings - Since you are representing 10x10 grids as 1D sequences, 1D relative positions may not capture this structure well. On the other hand, absolute position embeddings seem potentially problematic in the case of the SSTrfmr model, since they will not be consistently assigned to the same grid position if the text sequence is first and has varying length. Mitigating this may be important to provide for a fairer comparison with the SSTrfmr model.
* To what do you attribute the periodic loss spikes during training that are shown in Figure 4 (E)?
* I found the usage of "cross-attention" a bit confusing. For example, the single stream Transformer features cross-modal attention as an implicit consequence of self-attention over the concatenated sequence. I thought this would commonly be referred to as an instance of "cross-attention" between modalities.
* Does the dataset also contain visual renderings and natural language instructions to enable future work to study these tasks using familiar modalities?

---

> ### Author Response · Authors · 2023-11-21
> **Response to Reviewer a4Su (1/N)**
>
> We thank the Reviewer for their thorough and thoughtful feedback. Below, we have worked to address some of the weaknesses the Reviewer raised, particularly the strength of the baselines. We have included new experiments to directly address these concerns, taking into consideration this Reviewer’s suggestion. Below, we also address all questions directly.
>
> However, before addressing the Reviewer’s specific questions, we first wanted to address the two weaknesses the Reviewer raised. The first weakness was the lack of evaluation of deeper networks with more attention heads. We have now directly addressed this by performing additional experiments on a range of transformer layer depths (1, 2, 3, 4 layers) and attention heads (1, 4, and 8 heads), up to the size of BERT-small (4 layers, 8 heads; 12 model experiments in total). We have now included figures that detail the performance across this parameter sweep, with the primary conclusion that adding layers indeed aids with distractor and systematic generalization, but not productive generalization. We provide additional details below, and have incorporated the results for these new experiments in Figure 8 (in the Appendix), and have referenced these results in the manuscript's main text. In addition, we wanted to address another comment the Reviewer raised: “*I would have expected a reasonably sized Transformer model to achieve low training loss and reasonable IID generalization.*”
>
> We thank the Reviewer for their suggestion on expanding our evaluations. The Reviewer was correct in their intuition – deeper models help standard single stream transformers achieve improved IID generalization across all generalization splits (Fig. 8). We have now included the following text to the main manuscript (Results section 3.2) to reference these results: "(Note, however, that increasing depth (encoder layers) to Transformers improves IID generalization on these splits; Fig. 8)."
>
> The second weakness was the lack of clear distinction between our presented task, gCOG, and prior tasks such as the gSCAN task. We thank the Reviewer for this comment, and have now emphasized the primary distinctions between the two tasks. In brief, the two tasks require different neural network architectures. In terms of transformers, gSCAN is a generation task (requiring a decoder architecture) and gCOG is a classification task, which requires only an encoder Transformer. While models evaluated on gSCAN can incorporate encoder models to the architecture, as illustrated in one of the articles the Reviewer cites (Qiu et al., 2021), generating navigation instructions autoregressively adds complexity to studying compositional productivity, since autoregressive models are susceptible to exposure bias (Wang & Sennrich, 2020). Additionally, gCOG includes a distractor generalization split; our dataloaders are organized such that distractor generalization splits can interact with either systematic and/or productivity splits. We have now emphasized these differences in the manuscript, and have included citations to the studies mentioned by the Reviewer in the Related work sections.
>
> Related work revisions:
>
> “Our approach to constructing arbitrarily complex compositions of simple tasks is similar to (Ruis et al., 2020). However, it differs in three key ways. First, we focus on question-answer tasks (which require encoder-only architectures), rather than sequence-to-sequence learning tasks (which require decoder architectures, e.g., Qiu et al., 2021). Sequence decoding tasks introduce the added complexity of requiring autoregressive responses, which are susceptible to fundamental statistical challenges, such as exposure bias (Wang & Senn, 2021). Second, gCOG includes a distractor generalization split, in addition to systematic and productive compositional generalization splits. Finally, we methodically characterize different forms of generalization using simpler underlying abstractions (i.e., without the explicit use of image pixels).”
>
> We thank the reviewer for bringing to our attention Furer et al. (2021), and Shaw et al, (2021). References to these two studies are now included in the preceding paragraph of the above quoted passage.

---

> > ### Author Response · Authors · 2023-11-21
> > **Response to Reviewer a4Su (2/N)**
> >
> > We have additionally included discussion of Qiu et al., 2021 in the results section reporting that distractor generalization is improved using cross-modal attention: “While all models performed IID generalization well, only models that contained cross-attention mechanisms (CrossAttn and Perceiver models) exhibited excellent OOD distractor generalization (Fig. 3d). A related result was also reported in Qiu et al., (2021) using cross-modal self-attention.”
> >
> > Specific questions:
> >
> > *Why not try more layers and attention heads, e.g. following a standard hyperparameter setting for model size such as those of BERT-Base? Or even BERT-Small?*
> >
> > Thanks to the Reviewer’s suggestion, we have now extensively evaluated the task using a BERT-like model (similar to the SSTfmr) (Figure 8). In particular, we performed a parameter sweep, evaluating the effect of adding encoder layers (1, 2, 3, 4 layers) and attention heads (1, 4, 8 attention heads) on generalization performance. Indeed, we find that adding depth significantly aids in distractor and systematic generalization, while adding attention heads marginally improves performance on these splits. However, neither increasing layers nor attention heads improves productive generalization.
> >
> > We have adjusted the text throughout the manuscript to accommodate these new results. We consider these new results to complement the existing results on base neural network models, since experiments in the new Figure 8 evaluate models with greater complexity.
> >
> > Below is the caption for Figure 8; please see the updated PDF submission for the actual figure: “Figure 8: Evaluating generalization splits on BERT-like single-stream transformer models with varying layers and attention heads. We manipulate a generic encoder-only transformer based on the BERT architecture, evaluating the influence of the number of encoder layers (1, 2, 3, and 4 layers), and the number of attention heads per encoder layer (1, 4, and 8 heads). Overall, increasing layers improves generalization across distractor and systematic generalization, but not productive generalization. Increasing attention heads also marginally improves distractor and systematic generalization, but to a lesser extent than adding depth. A) Evaluation on distractor generalization across all model parameters. B) The effect of adding additional encoder layers on distractor generalization performance (averaged across all attention head configurations). C) The effect of adding attention heads on distractor generalization performance (averaged across all layer depth configurations. D-F) Evaluation on systematicity for depth 1 tasks (identical to generalization split in Fig. 4a). G-I) Evaluation on systematicity for depth 3 tasks (identical to generalization split in Fig. 4d). J-L) Evaluation on productivity split (identical to generalization split in Fig. 5a).”
> >
> > We have also revised the Contributions section (1.2) to reflect these new results: “2. A comprehensive evaluation of commonly-used base neural models (RNNs, GRUs, Transformers, Perceivers) on distractor, systematic, and productive generalization splits. We find that for distractor and systematic generalization, including a cross-attention mechanism across input modalities is important. However, all models fail on the productivity split. In addition, we include experiments demonstrating the impact of transformer depth and attention heads on all generalization splits in an encoder-only Transformer model (BERT-style models; Devlin et al. (2019))."
> >
> > Updates to the main manuscript (section 2.2, Model Architectures): “Finally, we report additional evaluations on deeper and larger SSTfmr models (i.e., BERT-style models) in Fig. 8 for all generalization splits. Those results demonstrate that improved distractor and systematic generalization performance can be achieved by adding parameters (i.e., increasing encoder depth and attention heads). However, adding parameters did not improve performance on productive compositional generalization splits.”
> >
> > *In Figure 2 (F) why does the single-stream Transformer have almost double the parameters of the double stream Transformer? For the other Transformers, do the encoder blocks used for the task vector and stimulus vector share parameters?*
> >
> > The reason the single-stream transformer has nearly double the parameters of the double stream transformer is due to the fact that attention is applied to all pairwise tokens in the SSTfmr, but attention is limited to within-modality attention (i.e., within rule tokens and within sensory tokens). Given that attention is an n^2 calculation on all pairs of tokens, this reason alone nearly doubles the number of parameters in the SSTfmr relative to the DSTfmr. All other architectural features are essentially fixed. For DSTfmr, CrossAttn model, and the Perceiver, there are separate encoder blocks for task vector and stimulus vectors.

---

> > > ### Author Response · Authors · 2023-11-21
> > > **Response to Reviewer a4Su (3/N)**
> > >
> > > *What optimizer and hyperparameters (e.g. learning rate) were used for training? How were these chosen? I didn't see these details in Appendix A.2.*
> > >
> > > We apologize that this was not more clearly presented in the Appendix A.2 – it was previously included under the description for each model. However, we have now included in the overview section of Appendix A.2: “All models were trained using the AdamW optimizer with a learning rate of 0.0001, and the loss was computed as the Cross Entropy between the target class and output vector.”
> > >
> > > *Position embeddings - Since you are representing 10x10 grids as 1D sequences, 1D relative positions may not capture this structure well. On the other hand, absolute position embeddings seem potentially problematic in the case of the SSTrfmr model, since they will not be consistently assigned to the same grid position if the text sequence is first and has varying length. Mitigating this may be important to provide for a fairer comparison with the SSTrfmr model.*
> > >
> > > We agree that a downside of the SSTfmr is that there is ambiguity in how different modalities interact with fixed or 1d positional encodings. This was part of the motivation for comparing that model with the DSTfmr and the CrossAttn models, where each modality would have its own set of positional encodings, and then lower-level features would then be integrated with either a shared MLP or cross-attention mechanisms.
> > >
> > > However, the additional experiments we performed in response to the Reviewer’s question 1 should demonstrate that adding more layers ameliorates the poor performance in the SSTfmr. Nevertheless, we agree that this is a potential issue for other problems that may arise, and could potentially be solved by either using separate positional encodings (for different modalities), and/or learnable positional encodings. However, we believe that this is beyond the scope of the current project, since adding additional encoder layers suffices for the systematic generalization split of this task.
> > >
> > > *To what do you attribute the periodic loss spikes during training that are shown in Figure 4 (E)?*
> > >
> > > The periodic loss spikes are due to a resampling of the training dataset. A resampling of the training dataset could have been due to reaching a checkpoint during model training, or a more pragmatic issue, such as disruption to a job/GPU. However, this resampling does not alter the overall distribution of the training set, and does not affect the conclusions of our study. We have now clarified this in the figures in which these loss spikes occur. Figure 4e and 5b caption: “(Note that the spikes in the loss function are due to a resampling of the training dataset due to model checkpointing and/or disruption to a compute job.)”
> > >
> > > *I found the usage of "cross-attention" a bit confusing. For example, the single stream Transformer features cross-modal attention as an implicit consequence of self-attention over the concatenated sequence. I thought this would commonly be referred to as an instance of "cross-attention" between modalities.*
> > >
> > > We appreciate the opportunity to clarify the distinction between **cross-modal self-attention** and **cross-attention**. Indeed, the Reviewer is correct that the SSTfmr theoretically implements cross-modal attention. However, this is applied through a distinct mechanism from “cross-attention”, as defined in (Jaegle et al., 2021). Critically, SSTfmr applies self-attention on multimodal inputs. In contrast, the CrossAttn models (Fig. 2d) applies cross-attention on two separate modalities, using the query of one modality, and the keys and values of the other. This implies that the attention matrix is not strictly n x n, but instead m x n, where m is for the number of tokens in the query matrix, and n is for the number of tokens in the keys and values matrix. This can significantly improve computational efficiency by reducing the need for a $n^2$ computation. We have clarified this in the Model Architectures section in the Appendix (A.2.1): “Since multimodal inputs were concatenated before being fed into the SSTfmr, self-attention was technically cross-modal. Note, however, that cross-modal self-attention is distinct from cross-attention, since self-attention always produces a square attention matrix, while cross-attention can produce a rectangular matrix, depending on the number of tokens in the query matrix and the number of tokens in the key and value matrices.”

---

> > > > ### Author Response · Authors · 2023-11-21
> > > > **Response to Reviewer a4Su (N/N)**
> > > >
> > > > *Does the dataset also contain visual renderings and natural language instructions to enable future work to study these tasks using familiar modalities?*
> > > >
> > > > Yes, thank you for the opportunity to clarify this feature of our task design. The dataset contains the ability to map categorical tokens to pixels (in the same way the task is visualized in Figure 1). The dataset also has a natural language translation, however, we note that currently, for task trees that are greater than depth 3, the natural language instructions could be grammatically incorrect. We are currently working on producing a syntax that will generate a multi-sentence syntax for task instructions that are appropriate for natural language processing. Revised text (in section 2.1): “Note, however, that for illustration purposes and interpretability, we maintain a visual and language representation in Figure 1 that is consistent with the original COG. Moreover, we additionally provide functionality in the dataset that allows the choice to generate data samples using either a categorical task encoding, or a task encoding with image pixels and natural language task instructions.”

---

> > > > > ### Comment · Reviewer_a4Su · 2023-11-21
> > > > >
> > > > > Thank you to the authors for including the new experiments on Transformers with >1 layer and >1 attention head, up to approximately "BERT small" sizes. And also thank you for clarifying the details of the available input formats released with the dataset. I think these changes have the potential to significantly improve the contribution of this work.
> > > > >
> > > > > However, I am a bit unsure about whether to increase my score. I still have issues with the current submission. I have no doubt that this paper could be revised in such a way that I would vote for acceptance, and that the dataset could be a nice contribution. But I think this requires significant enough changes to the experiments and paper that it would be more realistic to revise and resubmit this work to a future conference. Therefore, I will leave my score the same and let the AC take this into consideration.
> > > > >
> > > > > My main concern continues to be about the empirical experiments. I think the main results should be based on Transformers of a reasonable size and capacity, e.g. at least the "BERT small" size. The single layer Transformer results risk being misleading, since the expressiveness of such a small Transformer model is severely limited and may not correlate with the results of most Transformers used in practice.
> > > > >
> > > > > I also have some more minor concerns about the technical details:
> > > > >
> > > > > > Given that attention is an n^2 calculation on all pairs of tokens, this reason alone nearly doubles the number of parameters in the SSTfmr relative to the DSTfmr.
> > > > >
> > > > > I understand why this increases the amount of *computation*, but I don't understand why this increases the number of *parameters*, as attention parameters are shared along the sequence dimension.
> > > > >
> > > > > I also recommend better understanding the loss spikes observed during training, and tuning of the learning rate if models continue to underfit the training data.

---

> > > > > > ### Author Response · Authors · 2023-11-21
> > > > > > **2nd Response to Comment by Reviewer a4SU**
> > > > > >
> > > > > > We thank Reviewer a4Su for their prompt response. We understand their hesitation that the single layer Transformer results may be perceived as misleading. Thus, we want to emphasize some key points and some revisions that we hope will mitigate any potential confusions.
> > > > > >
> > > > > > First, however, we would like to emphasize that we think that the failure mode of the single layer Transformer is informative. This is because it provides critical information on how alterations to this base Transformer architecture (i.e., increasing encoder layers or attention heads) can improve systematic and distractor generalization. However, we agree with the Reviewer that without additional context (i.e., the results on deeper single stream transformers), these results can be misleading. Thus, we will more strongly emphasize the new experiments that evaluate BERT-small sized models by moving them from the Appendix to the main text. These changes will make it clear that either cross-attention between modalities, **or inclusion of deeper single stream Transformers**, can improve systematic and distractor generalization. Importantly, we will include a new Results section (in the main text) titled, “*Impact of layer depth and attention heads in SSTfmrs on generalization*”. In this section, we will describe that standard BERT-small sized models can indeed perform multimodal systematic and distractor generalization, but not productive compositional generalization. To make space for this revision, we will move the final section, “Representation analysis of model architectures” to the Appendix.
> > > > > >
> > > > > > We believe that these changes will ensure that there will be no confusion about the limitations of the capacity for single stream Transformer models. For example, we have now revised the abstract (and will continue revising the manuscript with the proposed changes above) to reflect these key findings, and to mitigate any possibility that the single layer transformer results would be deemed misleading.
> > > > > >
> > > > > > Abstract: “... We found that across model architectures (e.g., RNNs, Transformers, Perceivers, etc.), **models with multiple attention layers**, or models that leveraged cross-attention mechanisms between input domains, fared better. Our positive results demonstrate that for multimodal distractor and systematic generalization, either cross-modal attention **or models with deeper attention layers** are key architectural features required to integrate multimodal inputs. On the other hand, neither of these architectural features led to productive generalization, suggesting fundamental limitations of existing architectures for specific types of multimodal generalization.
> > > > > >
> > > > > > **Regarding the comment**: *I understand why this increases the amount of computation, but I don't understand why this increases the number of parameters, as attention parameters are shared along the sequence dimension.*
> > > > > >
> > > > > > We apologize for the confusion on our behalf -- the reviewer is correct. The confusion stems from a mislabeling of the xtick axis in Figure 2F. DSTfmr and SSTfmr were inadvertently swapped, and this has been corrected. We have reviewed the remainder of the code responsible for generating figures and ensured that the xtick labels have been labeled correctly. DSTfmr has more parameters due to having double the MLP layers (one for each encoder block).
> > > > > >
> > > > > > **Regarding the comment**: * I also recommend better understanding the loss spikes observed during training, and tuning of the learning rate if models continue to underfit the training data.*
> > > > > >
> > > > > > In prior experiments that we did not report in the manuscript, the loss spikes during training have less to do with underfitting of the training data, but more to do either 1) The size of the overall set of possible stimulus attributes. Specifically, if there are 10 colors, 26 letters, and 100 locations, there are 26,000 possible object combinations. In prior experiments where we include fewer object combinations, the fewer the total possible of combinations, the smaller the loss spikes are from data resampling.

---

### Official Review · Reviewer_DJb6 · 2023-11-08

**Soundness:** 3 good
**Presentation:** 3 good
**Contribution:** 3 good
**Rating:** 8
**Confidence:** 4

**Summary:**

The paper introduces a new multimodal question answering benchmark for out-of-distribution generalization, specifically covering task compositionality, robustness to distractors and combinatorial generalization. It uses this benchmark to evaluate various models and analyze their performance.

**Strengths:**

- **Topic**: The paper studies an important topic which in my opinion is underexplored in current deep learning research. Especially given the tendency these days to scale training up to vast amounts of data, I believe it is particularly important to design carefully controlled benchmarks that can: evaluate the model’s performance from a critical and cautious standpoint, point to their fundamental limitations (e.g. systematic generalization), and support further research about ways to overcome these.
- **Evaluation**: The paper offers both extensive extrinsic evaluation, with performance comparison of various models on the different generalization skills, as well as intrinsic analysis of their internal representations’ degree of alignment to the stimuli.
- **Clarity**: The writing quality is good and the paper is clear and easy to follow. The paper is well-organized, claims and findings are clearly stated, and useful figures and diagrams are provided.
- **Related Works**: It does a good job in providing the relevant context, motivation and related works.
- **Contribution**: The empirical findings of the paper on the benefits and limitations of different inductive biases such as recurrent and attention-based are important and may be of broad interest to the community.

**Weaknesses:**

- **Pre-trained models** The paper focuses on models trained from scratch rather than pre-trained. This could be a strength and a weakness. On the one hand, it allows for isolating the contribution of the architectural choices from other factors of optimization, and training data. On the other hand, it has been observed that by training models at large enough scales enables the emergence of generalization capabilities, which we don’t see in smaller scales. I think it will be critical to also analyze the performance of pretrained models on the benchmark, in order to strengthen the paper.
- **Visual Simplicity**: The visual side of the benchmark is quite rudimentary, featuring colorful letters. Extending it to a larger range of visual tokens/objects, that could have more than one property (color), and a broader set of elements and variations (than 26 letters), could be a straightforward extension that could help make it a bit more challenging visually.

**Questions:**

- **COG task**: It will be useful to discuss the COG task (rather than just mentioning it) before describing the new gCOG one, so that it will be clearer to the reader what are new contributions of the new benchmark compared to COG and the degree of their importance. In the overview diagram I would also recommend showing a sample also from COG to make the differences clearer.
- **Grid size / generalization**: It could be interesting to vary the size of the grid in training/evaluation and study its impact on model’s performance.
- **Terminology**: I recommend changing the phrase “Distractor generalization” to one that better conveys it’s about changing the answer distribution. Maybe e.g. answer distribution shift. I also recommend changing the name “Systematic compositional generalization” to “combinatorial generalization”, to emphasize that the main point is the generalization to permutation, and also to better contrast it with the following “Productive generalization” (which could also be systematic).
- **Figures**: Would be good to increase the size of the plots in Figure 3b. It will also be good the increase the distance and visual separation between the sub-figures in each figure throughout the paper.
- In the introduction: “multimodal question-answer” -> “answering”.
- “This design allowed us” -> “This design allow us”.

---

> ### Author Response · Authors · 2023-11-21
> **Response to Reviewer Djb6 (1/N)**
>
> We thank Reviewer Djb6 for their thoughtful review, and their positive assessment of our manuscript. Below we address the weaknesses the Reviewer raised, and directly respond to questions.
>
> Weakness 1: Lack of evaluation using pre-trained models. We agree with the Reviewer that there is utility in assessing how a pretrained model performs on a new task to the literature. However, when trying to address this question regarding our specific task, we realized that most models would not be able to perform this task out-of-the-box without any fine-tuning, since tokens in the current task set up have no intrinsic meaning. Since we could not directly test mainstream pretrained transformer models (without devising a specific way of first aligning domains through finetuning), we addressed a related question raised by the Reviewer: How would mainstream architectures (such as BERT-like models; Devlin et al., 2019) fare on our benchmark? Moreover, much of the motivation for this manuscript was to focus on questions such as: What architectural components are essential for distractor, systematic, and productive generalization? To this end, we performed additional experiments to evaluate how standard architectures – such as BERT-like architectures -- generalize appropriately. These experiments focused on investigating how increasing encoder-layer depth, and increasing attention heads in each encoder layer, influenced generalization performance. These results have now been included as Figure 8 in the Appendix. Nevertheless, we agree that evaluating the out-of-the-box performance of pretrained models on a pixel and natural language variant of this task is important for future work to explore (once a sensible fine-tuning procedure to align the two is agreed upon).
>
> Weakness 2: Visual Simplicity. We agree with this sentiment; many of the visual stimuli are indeed rudimentary. However, note that in this task, we will deploy a config file that allows one to extend the number and choice of symbols that can be included. Moreover, the mapping from categorical tokens to image pixels implies that any symbol with a UTF-8 encoding can be easily incorporated (e.g., letters, numbers, symbols). (In principle emojis can also be included, but note that the mapping emojis to a specific color attribute is not straightforward.) Release of the dataset will include the standard splits that we have included with this manuscript, in addition to datasets that map categorical tokens to image pixels and natural language. We will also provide the raw dataloaders that can be customized as desired. We have clarified this in the Experimental Design section (2.1): “The total number of unique individual tasks (i.e., task trees of depth 1) is 8 operators ∗ 26 shapes ∗ 10 colors = 2080 unique individual task commands, but can be straightforwardly extended with modifications to the configuration file.”
>
> ## Specific questions. (*Italics* indicates Reviewer comment/question.)
>
> *COG task: It will be useful to discuss the COG task (rather than just mentioning it) before describing the new gCOG one, so that it will be clearer to the reader what are new contributions of the new benchmark compared to COG and the degree of their importance. In the overview diagram I would also recommend showing a sample also from COG to make the differences clearer.*
>
> We thank the Reviewer for the suggestion. We have now clarified the explicit differences between COG and gCOG in the Experimental Design section (2.1): “gCOG is a configurable question-answer dataset, originally inspired from COG (Yang et al., 2018), that programmatically composes task instructions, and then generates synthetic stimuli to satisfy those instructions on-the-fly (Fig. 1). The primary modifications in gCOG are 1) differences in the set of task operators, 2) the ability to use categorical tokens to allow for generic testing of multimodal reasoning, and 3) the ability to allow for arbitrarily long task trees to assess productive compositional generalization, in addition to distractor and systematic generalization (e.g., see Appendix Fig. 7). Importantly, the original COG task did not allow for tasks with more than a single conditional statement, e.g., a task tree of depth 3, making it ill-suited to evaluate productive compositional generalization... We additionally provide functionality in the dataset that allows the choice to load samples using either a categorical task encoding, or a task encoding with image pixels and natural language task instructions."
>
> Due to copyright and the license of the original COG paper, we were unable to include a figure from the original COG task/paper (Springer publishing). However, we have included sample questions/queries from the COG task in the Appendix: “A few example queries from the original COG task include: `What is the color of the latest triangle? Point to the latest red object. If a square exists, then point to the current x, otherwise point to the last b.’”

---

> ### Author Response · Authors · 2023-11-21
> **Response to Reviewer Djb6 (N/N)**
>
> *Grid size / generalization: It could be interesting to vary the size of the grid in training/evaluation and study its impact on model’s performance.*
>
> This is a good suggestion that we originally considered. We will provide a configuration file that specifies the dimensions of the task grid, so altering the specific grid size can be straightforwardly implemented in future studies.
>
> *Terminology: I recommend changing the phrase “Distractor generalization” to one that better conveys it’s about changing the answer distribution. Maybe e.g. answer distribution shift. I also recommend changing the name “Systematic compositional generalization” to “combinatorial generalization”, to emphasize that the main point is the generalization to permutation, and also to better contrast it with the following “Productive generalization” (which could also be systematic).*
>
> On distractor generalization: Initially, we planned to call it noise generalization, as distractors can be viewed from the perspective of noise. However, we settled on ‘distractor’ generalization, as it is similar to how distractors are commonly referred to in psychology tasks. In this scenario, distractors serve to make it harder to evaluate the task instruction due to additional visual distractors. Adding/subtracting distractors would not change the distribution of the answer.
>
> On systematic and productive compositional generalization: We have opted to keep the terms systematic and productive compositional generalization, given the established precedence in the machine learning literature for those terms (see Hupkes et al., 2020, JAIR, for a comprehensive review defining systematic and productive compositionality). However, the Reviewer is correct that systematic compositional generalization is intuitively similar to combinatorial generalization, since a novel combination of task instructions is employed. We have therefore made this link in the Introduction, when we first introduce the term systematic compositional generalization. Revisions to the Introduction: “This design allowed us to comprehensively evaluate a variety of neural network architectures on tasks that test for three different forms of OOD generalization: 1) Distractor generalization (generalization in the presence of a different noise distribution), 2) Systematic compositional generalization (generalization to new permutations of task structures, **i.e., combinatorial generalization)**, and 3) Productive compositional generalization (generalization to task structures of greater complexity).”
>
> *Figures: Would be good to increase the size of the plots in Figure 3b. It will also be good the increase the distance and visual separation between the sub-figures in each figure throughout the paper.*
>
> We have now increased the size of the plots in Figure 3b, splitting panel 3b in to 3b and 3c. We have also worked to increase the amount of visual separation between panels in Figure 2, but note for some figures this was difficult due to space constraints.

---

> > ### Comment · Reviewer_DJb6 · 2023-12-04
> > **Thanks!**
> >
> > Dear authors, thank you for the responses to each of my comments, about pre-trained models evaluation, visual simplicity, COG, terminology and figures, I find them very useful!

---

### Author Response · Authors · 2023-11-21
**Summary of revisions and overall response to Reviewers**

We thank all the Reviewers for their thoughtful evaluation of our manuscript. We appreciate that Reviewers DJb6 and skmj found our paper comprehensive, and that Reviewers a4SU and skmj found it well-written and easy to read. In replies to each Reviewer’s comments, we directly address the weaknesses and specific questions for each Reviewer. However, we would like to highlight two primary revisions we have implemented that we believe significantly improve our manuscript. Importantly, these two revisions improve the model evaluation on our experimental design/task, as well as further clarify the significance and utility of the gCOG dataset.



1. New experiments on larger single stream transformer models with more layers and attention heads (BERT-like models). To assess the role of Transformer size and parameters (encoder layers and attention heads), we performed experiments to systematically evaluate the effect of increasing the number of Transformer layers and attention heads per layer on all generalization splits (distractor, systematic, and productive generalization). We experimented on Transformer encoder models up to the size of BERT-small, including models with 1, 2, 3, and 4 layers, and 1, 4, and 8 attention heads, totaling 12 new models in total. In brief, the primary conclusion from these experiments is that adding layers helps with distractor and systematic generalization, but not productive generalization. We have adjusted the text throughout the manuscript to accommodate these new results, including a new figure (Fig. 8, Appendix) that detail these new results. We consider these new results to complement the existing results on base neural network models, since these experiments evaluate models with greater complexity. Appendix Section A.2.1 provides additional details regarding these models.

2. A detailed description of the novel features of the dataset, and comparison with existing benchmarks. We have clarified in the response, as well as in the manuscript, that the dataset – one of the primary contributions of this study – includes both task encodings with categorical tokens, and task encodings with image pixels and natural language instructions. This will provide researchers with resources to study multimodal reasoning using either generic tokens (for theoretical studies) or in more applied and natural settings, e.g., multimodal vision/language.

We hope that our revised manuscript will be valuable to the broader community, and inspire further research into multimodal compositional generalization and reasoning, which remains a challenge for modern AI systems.

---

### Meta-Review · Area_Chair_Ze7i · 2023-11-28

**Metareview:**

This paper introduces a new dataset that examines multimodal generalization on transformers. New and careful datasets to examine model performance are a useful contribution. The authors also contribute some experiments what which kinds of transformer properties yield different kinds of generalization behavior, they have worked toward extending these experiments to a more diverse set of models (partially addressed in rebuttal)  -- with these included in a revised manuscript, it's worth presenting as a poster.

**Justification For Why Not Higher Score:**

Extending these experiments to a more diverse set of models (partially addressed in rebuttal) including pretrained ones would strengthen the paper.

**Justification For Why Not Lower Score:**

New and careful datasets to examine model performance are a useful contribution on their own, the authors also contribute some experiments what which kinds of transformer properties yield different kinds of behavior

---

### Decision · Program_Chairs · 2024-01-16

Accept (poster)